# Mitochondrial ROS Produced in Human Colon Carcinoma Associated with Cell Survival via Autophagy

**DOI:** 10.3390/cancers14081883

**Published:** 2022-04-08

**Authors:** Eun Ji Gwak, Dasol Kim, Hui-Yun Hwang, Ho Jeong Kwon

**Affiliations:** Chemical Genomics Laboratory, Department of Biotechnology, College of Life Science and Biotechnology, Yonsei University, Seoul 03722, Korea; ejgwak37@yonsei.ac.kr (E.J.G.); rlaekthf0123@yonsei.ac.kr (D.K.); ghkdgmldbs@yonsei.ac.kr (H.-Y.H.)

**Keywords:** UQCRB, mROS, autophagy, colorectal cancer, lysosome

## Abstract

**Simple Summary:**

Human colon carcinoma remains one of the major causes of cancer-related death worldwide. Ubiquinol cytochrome c reductase binding protein (UQCRB) has been reported as a biomarker of colorectal cancer, but its role in tumor growth has not been clarified. CRC cells often exhibit high autophagic flux under nutrient deprivation or hypoxic condition and increased autophagy activation in cancer cells involving the recycling of cell components to facilitate survival in a tumor microenvironment. Here we show that UQCRB is overexpressed in HCT116 cells compared to CCD18co, normal colon fibroblast cells. Mechanistically, the increasing level of mitochondrial ROS (mROS) caused by UQCRB overexpression can release Ca^2+^ by the activation of the lysosomal transient receptor potential mucolipin 1 channels. This activation triggers transcription factor EB nuclear translocation and lysosome biogenesis leading to autophagy flux. Collectively, we identified that the increasing level of mROS by the overexpression of UQCRB in human colon carcinoma could link to autophagy for colorectal cancer survival. These results lead to a translational impact that a UQCRB inhibitor could be a potential anticancer agent for human colon carcinoma treatment.

**Abstract:**

Human colon carcinomas, including HCT116 cells, often exhibit high autophagic flux under nutrient deprivation or hypoxic conditions. Mitochondrial ROS (mROS) is known as a ‘molecular switch’ for regulating the autophagic pathway, which is critical for directing cancer cell survival or death. In early tumorigenesis, autophagy plays important roles in maintaining cellular homeostasis and contributes to tumor growth. However, the relationships between mROS and the autophagic capacities of HCT116 cells are poorly understood. Ubiquinol cytochrome c reductase binding protein (UQCRB) has been reported as a biomarker of colorectal cancer, but its role in tumor growth has not been clarified. Here, we showed that UQCRB is overexpressed in HCT116 cells compared to CCD18co cells, a normal colon fibroblast cell line. Pharmacological inhibition of UQCRB reduced mROS levels, autophagic flux, and the growth of HCT116 tumors in a xenograft mouse model. We further investigated mutant UQCRB-overexpressing cell lines to identify functional links in UQCRB-mROS-autophagy. Notably, an increasing level of mROS caused by UQCRB overexpression released Ca^2+^ by the activation of lysosomal transient receptor potential mucolipin 1 (TRPML1) channels. This activation induced transcription factor EB (TFEB) nuclear translocation and lysosome biogenesis, leading to autophagy flux. Collectively, our study showed that increasing levels of mROS caused by the overexpression of UQCRB in human colon carcinoma HCT116 cells could be linked to autophagy for cell survival.

## 1. Introduction

Mitochondria contribute to the malignant transformation of normal cells into neoplastic precursors by inducing the production of reactive oxygen species (ROS) [1]. Complex I and complex III of the electron transport chain have been recognized as the sites of superoxide production [2]. The mitochondrial ROS (mROS) can cause oncogenic DNA defects to accumulate and activate oncogenic signaling pathways [1]. Initially, mROS studies focused on the damaging effects of mROS, but recently, research has been shifting to determine whether mROS can act as a signaling molecule to support pro-growth responses in cells [3]. Targeting mitochondria-to-cell redox communication can alter activation of the cancer-promoting signaling pathway, thereby interfering in mitochondria-released oxidant signaling events, which could be a promising cancer therapy [4].

Ubiquinol-cytochrome c reductase-binding protein (UQCRB) is a 13.4-kDa subunit of complex III in the mitochondrial respiratory chain [5] shown to play an important role in transporting electrons and maintaining the structure of complex III [6]. UQCRB mutation can cause mitochondrial defects and is associated with several diseases. A case report of a Turkish female infant showed that 4-bp deletion in the UQCRB encoding gene resulted in metabolic disorders, such as hypoglycemia and lactic acidosis [7]. In a previous study, we constructed a mutant UQCRB-expressing stable cell line (MT). Notably, MT cells exhibit more cell growth and pro-angiogenesis activities than the wild type of HEK293 cells [8]. In addition, MT cells’ mitochondria showed morphological abnormalities, and they were more sensitive to UQCRB inhibitors terpestacin and A1938 than HEK293 [8,9]. These results collectively show that the redox stress mechanism that operates through increasing mROS generation can cause pseudo-hypoxia. Thus, it may link mitochondrial abnormalities to angiogenesis-related diseases and cancer [10].

Meanwhile, in our previous study, we discovered UQCRB as a target protein of terpestacin, a small molecule that inhibited angiogenic responses. Terpestacin is a sesterterpene natural product that is produced by *Embellisia chlamydospora* [5]. Based on this result, we studied a target-based screen with structural information on the binding mode of terpestacin and UQCRB. Subsequently, the synthetic compound A1938 was generated and identified to exhibit potent antiangiogenic activity both in vitro and in vivo by directly binding to UQCRB [11].

Despite advances in diagnosis and treatment, colorectal cancer (CRC) remains one of the major causes of cancer-related death in both men and women worldwide. In 2020, there were over 1 million new cases of CRC and over ~500,000 related deaths worldwide [12]. The relationship between UQCRB and CRC has been extensively studied. The *UQCRB* gene and protein levels are found at higher levels in CRC tissues than in adjacent non-tumor tissues [13]. In addition, UQCRB is more frequently highly expressed in CRC patients whose diseases are more advanced and who have poor pathological differentiation. The expression level of UQCRB-related circulating miR-4435, which regulates the invasion of HCT116 cells and migration functions, is upregulated in HCT116 [9,13]. These results demonstrate that UQCRB expression is correlated with CRC progression, making it a biomarker for diagnosing CRC [9,13].

Cancer cells uniquely reprogram their cellular activities to support rapid proliferation and migration and counteract metabolic and genotoxic stresses during cancer progression. In a previous study, we reported the rapid proliferation and invasion ability of mutant UQCRB-expressing cells [8]. In the present study, we found that UQCRB expression was significantly higher in HCT116 cells than in normal CCD18co cells. Moreover, MT cells expressed a higher amount of autophagy marker protein than normal HEK293 cells, which contributed to the survival of the MT cells. Furthermore, autophagy and lysosomal activity induced by UQCRB overexpression were abolished by the UQCRB inhibitor, A1938. Therefore, targeting mitochondrial UQCRB using the UQCRB inhibitor could be a possible treatment for CRC patients.

## 2. Results

### 2.1. The UQCRB-Overexpressing CRC Cell Line HCT116 Is Sensitized by A1938-Induced Cell Death

UQCRB was discovered as a target protein of the anti-angiogenic natural small molecule terpestacin [5]. Overexpression of UQCRB causes the production of mROS, and inhibition of UQCRB decreases mROS generation in various cancer cell lines, including HT1080, HepG2, and U87MG [5,11,14,15]. Although the relationship between UQCRB and CRC has been studied [13], the underlying mechanism of the inhibition of UQCRB-induced cell death remains unknown. To explore the function of UQCRB in CRC, we investigated UQCRB expression levels in HCT116 cells by Western blot analysis using anti-UQCRB antibodies and found that endogenous UQCRB levels in CRC cells were higher than those in normal CCD18Co cells (Figure 1A). Because the basal level of ROS has been shown to be increased in cancer cells when compared with normal cells [16], we investigated whether the inhibition of UQCRB regulates ROS generation in HCT116 cells. Notably, the high mROS levels in HCT116 cells were suppressed by treatment with A1938, a UQCRB inhibitor (Figure 1B,C).

Prior to characterizing the effect of the underlying mechanism of UQCRB inhibition activity, we determined the effects of A1938 on cell proliferation and viability of HCT116 cells. The HCT116 cells were treated with various doses of A1938 for 0–72 h, and their cell growth was measured using a 3-(4,5-dimethylthiazol-2-yl)-2,5-diphenyltetrazolium bromide (MTT) colorimetric assay. The results showed that various doses of A1938 affected the growth of HCT116 cells, with an effective concentration starting at 30 μM (Figure 1D). In addition, trypan blue staining showed that HCT116 cells treated with A1938 in a medium containing 10% serum did not show cytotoxicity up to 50 μM (Figure 1E). A concentration of 30 μM A1938 was therefore used in subsequent studies.

Several types of CRC-derived cells, including HCT116 cells, often exhibit high autophagic flux under nutrient deprivation or hypoxia [17,18,19,20]. For these reasons, we investigated whether the UQCRB inhibitor A1938 could affect the growth of cancer cells by inhibiting autophagy. At doses above 20 μM under serum-starved conditions, A1938 had a cytotoxic effect on HCT116 cells of over 67%. In addition, when HCT116 cells were treated with 30 μM A1938 in a medium containing 10% serum, cell viability was about 90%. However, at doses of 30 μM under serum-starved conditions, cell viability was decreased below 13% (Figure 1E). HCT116 cells were treated with the autophagy inhibitor chloroquine (CQ) (Figure 1F) to determine if the decreased cell growth and viability were the results of autophagy inhibition. Co-treatment of CQ and A1938 showed a further reduction in cell viability, indicating that autophagy inhibition by CQ and A1938 could play an additive role in these cells. Increased autophagy activation in HCT116 cells involves recycling cell components to facilitate survival in a nutrient-deprived environment. However, inhibition of autophagy flux in HCT116 cells under serum starvation conditions would prevent autophagic recycling of cell components to allow survival. Taken together, these results suggested that inhibiting autophagy flux by CQ in HCT116 cells led to further increased cytotoxicity, and it could be associated with UQCRB inhibition in the nutrient starvation condition.

### 2.2. Autophagy Is Induced in Mutant UQCRB-Overexpressing Cells (MT)

We further investigated the autophagic function of UQCRB with the genetically increased conditions using HEK293 cell lines with normal and mutant UQCRB expressing cells. Previously, Chang et al. constructed the mutant UQCRB expressing HEK293 cells (MT), and the overexpression of mutant UQCRB causes the rapid production of mROS and abnormal mitochondrial morphology, which is characterized by significant swelling [8]. We, therefore, determined the effect of mROS production caused by UQCRB overexpression on autophagy during cell proliferation. First, we examined whether the MT stable cell line retained its characteristics. Immunoblotting using anti-UQCRB and anti-Myc antibodies showed that validated MT cells overexpressed UQCRB, which is consistent with the results of a previous study (Figure 2A). Furthermore, mROS levels were higher in MT cells than in control HEK293 cells (Appendix A), which suggested that the production of large amounts of mROS in MT cells induced autophagy in a similar way as treatment with exogenous oxidants.

Next, we confirmed the presence of autophagy markers LC3 and p62 in MT cells using immunoblotting. During autophagy, LC3-I is converted to LC3-II, which is a key component of autophagosomes. In ubiquitin-dependent autophagy, autophagic cargo receptor proteins such as p62 act as degradation substrates of the autophagic membrane so that p62 protein levels can be increased while autophagy occurs [21]. When treated with serum and starved, MT cells had higher LC3-II and p62 levels than normal HEK293 cells (Figure 2B). Next, autophagy induction was examined by immunocytochemistry using LC3 antibodies. HEK293 and MT cells were incubated in SF medium for 24 h, after which LC3 puncta were only observed in MT cells (Figure 2C). This result demonstrated that UQCRB overexpression contributed to mROS production, making cells sensitive to serum-free, nutrient-deprivation environments that lead to autophagy.

Autophagy is important to the constitutive turnover of intracellular components and for the elimination of potentially damaging or abnormal intracellular components [22]. To investigate autophagic flux, CQ, an inhibitor of autophagy that impairs autophagosome fusion with lysosomes, was used to treat HEK293 and MT cells [23]. First, HEK293 and MT cells were treated with 5 μM of CQ for 0–6 h to investigate their autophagy and autophagic flux kinetics. In MT cells, LC3B and p62 increased in a time-dependent manner, whereas p62 protein levels did not change or even decrease over 4 h in CQ-treated HEK293 cells (Figure 2D). However, if autophagic flux is occurring, the amount of LC3-II will be higher in the presence of the inhibitor [24]. CQ treatment blocked autophagosome degradation, resulting in higher levels of LC3-II in both HEK293 and MT cells. However, the densitometric values of LC3-II and LC3-I indicated that MT cells exhibited a greater LC3-II conversion rate than control cells (Figure 2E). In addition, after treatment with CQ for 6 h, p62 levels in MT cells were significantly higher than those in HEK293 cells. To further validate the effect of UQCRB on autophagy flux, a double-tagged mRFP-GFP-LC3 plasmid was used to visualize the transition from neutral autophagosomes to acidic autolysosomes based on the different pH stabilities of mRFP-LC3 and GFP-LC3. GFP fluorescence is unstable in acidic compartments, whereas mRFP fluorescence is relatively stable, even in the acidic environment of lysosomes [25]. Accordingly, autophagic flux was monitored according to the decrease in co-localization of green and red fluorescence, which appeared yellow to indicate the presence of autophagosomes, and an increase in red fluorescence, which reflects the presence of autolysosomes. An average of 25 red puncta were detected in HEK293 cells, while an average of 56 were detected in MT cells (Figure 2F). Together, these results indicated that the normal expression of UQCRB could not actively induce autophagy, suggesting that the overexpression of UQCRB induced more autophagy flux.

### 2.3. The Effects of UQCRB Overexpression on Activation of TRPML1 Ca^2+^ Channels and TFEB via mROS

The mROS is known to directly and specifically activate the lysosomal Ca^2+^ channel, TRPML1, inducing lysosomal Ca^2+^ release that activates calcineurin [26]. We hypothesized that increasing mROS levels by overexpressing UQCRB could trigger lysosomal TRPML1 channels and induce lysosomal Ca^2+^ release. HEK293 and MT cells were therefore transfected with GCaMP3-ML1 encoding a lysosome-targeted Ca^2+^ probe. The structure of the GFP tag on the channel protein in the outer lysosomal membrane shifts when this Ca^2+^ probe captures Ca^2+^ ions [27]. MT cells released higher Ca^2+^ levels than HEK293 cells without any other treatment. In addition, the TRPML1 channel agonist, ML-SA1, induced more Ca^2+^ release in MT cells than in HEK293 cells. Accordingly, increased mROS levels in MT cells resulted in the activation of lysosomal TRPML1 more frequently by ML-SA1 (Figure 3A,B), showing that overexpression of UQCRB indirectly regulated the lysosomal Ca^2+^ ion channel, causing Ca^2+^ to be released from lysosomes into the cytosol.

TFEB-nuclear translocation can be stimulated by the release of lysosomal Ca^2+^ and the Ca^2+^-dependent phosphatase, calcineurin [27]. TFEB is normally maintained in an inactive state by phosphorylation in the cytosol. However, TFEB translocation into the nucleus can activate autophagy- and lysosome-related genes as a transcription factor [28]. Thus, TFEB localization could be physiologically related to TFEB activation. In addition, ROS levels are elevated during starvation [29], which activates TFEB and promotes autophagy [27].

To investigate whether TFEB translocation was related to UQCRB in affecting mROS levels, the enhanced green fluorescence protein (EGFP)-TFEB plasmid was transfected into HEK293 and MT cells to observe the nuclear translocation of TFEB directly. When treated with serum, TFEB translocated into the nucleus in 3 out of 20 HEK293 cells (15%) but into 19 of 37 MT cells (51%) (Figure 3C,D).

Previously, the effect of A1938 on the proliferation of HEK293 and MT was investigated by our group. A1938 treatment inhibited the proliferation of HEK293 more weakly than that of MT cells, demonstrating no cell toxicity to HEK293 and more sensitive to the MT cells [8]. A1938 treatment in MT cells reduced the mROS production levels (Appendix A). After A1938 treatment, the nucleus–cytoplasm ratio of TFEB was 35% lower than in MT cells treated with dimethyl sulfoxide (DMSO) (Figure 3C,D). Collectively, these results demonstrated that UQCRB inhibition reduced mROS production through the regulation of lysosomal Ca^2+^-mediated TFEB signaling.

### 2.4. The Effect of A1938 on Increased Lysosome Activity in MT Cells

The role of lysosomes in degradation and recycling involves a cellular housekeeping function [27]. Intracellular substrates are delivered to the lysosome by the autophagic pathway by fusing autophagosomes with lysosomes. TFEB is a master regulator of lysosomal and autophagic functions and energy metabolism [27]. Thus, we examined lysosome biogenesis and activity using LysoTracker Red (LTR) and acridine orange (AO) to stain acidic compartments, such as lysosomes.

After LTR staining, red puncta showed the locations of lysosomes [30]. DMSO-treated MT cells had higher levels of acidic vesicles than DMSO-treated normal HEK293 cells. Specifically, an average of nine red puncta were detected in HEK293 cells, while an average of 39 were detected in MT cells. A1938 treatment reduced the number of red puncta to an average of 17 in MT cells (Figure 4A,B).

Lysosomes play a role in cell component degradation by maintaining an acidic environment with a pH of approximately 5 to facilitate the functioning of hydrolytic enzymes. Treating MT cells with A1938 caused their AO fluorescence intensity to significantly decrease, which indicates that lysosome function was inhibited (Figure 4C).

Because TFEB is a transcriptional regulator of lysosome biogenesis [31], we investigated how UQCRB was related to mROS during lysosome biogenesis. LAMP1 is used as a lysosomal marker, and LAMP1-positive organelles are considered as lysosomal compartments [32]. The expression of the lysosomal protein LAMP1 was higher in MT cells than in HEK293 cells, and A1938 treatment decreased lysosomal biogenesis (Figure 4D,E). These results suggest that an increased level of mROS produced by UQCRB overexpression induced lysosomal biogenesis through TRPML1-TFEB activation and that this process was regulated by a UQCRB inhibitor.

### 2.5. The Effect of A1938 on Autophagy Flux by Suppression of mROS Generation

ROS can function as a signaling molecule that induces autophagy, which tends to degrade dysfunctional mitochondria. Zhang et al. evaluated the effect of mROS on autophagy induction using the exogenous oxidants carbonyl cyanide *m*-chlorophenylhydrazone and rotenone [26]. We, therefore, evaluated the effect of A1938 on autophagy by suppressing mROS production in MT cells. MT cells treated with A1938 exhibited increased LC3B-II and p62 protein levels when compared with MT cells treated with DMSO (Figure 5A). In cells expressing mRFP-GFP-LC3, A1938 treatment resulted in increased yellow fluorescence in MT cells (Figure 5B,C). Similar to A1938, treatment of mROS scavenger, Mito-Tempo (mTP) increased yellow fluorescence in MT cells as well. By the result of Figure 4, we suggest A1938 decreases the activity of lysosomes in MT cells and indicating that inhibiting the production of mROS suppressed autophagy activation (Appendix A). Overall, autophagosomes likely accumulated due to inhibition of autophagic degradation, which indicates that A1938 decreased autophagy flux. 

### 2.6. HCT116 Is Sensitized by A1938-Induced Autophagy Regulation

Cancer cells are usually able to tolerate stressed environments through autophagy, which removes ROS and damaged proteins and maintains mitochondrial functioning and cellular metabolism, to promote survival under stress [16]. As shown in Figure 1, A1938-induced cell death was accompanied by serum-free, nutrient-deprived environmental conditions in HCT116 cells. To confirm that A1938-mediated autophagy inhibition affected HCT116 survival under serum starvation conditions, treatment with A1938 and mTP in serum-free conditions for 24 h resulted in increased yellow fluorescence in HCT116 cells indicating that inhibiting the production of mROS suppressed autophagy flux (Figure 6A,B). In addition, HCT116 cells were treated with A1938 for 24 h under starvation conditions, which dramatically increased their LC3B levels (Figure 6C). These results indicate that A1938 inhibited autophagy flux in HCT116 by decreasing the expression of mROS.

In this study, the inhibition of UQCRB protein by A1938 reduced autophagy and lysosomal activity in MT cells (Figure 4 and Figure 5). We, therefore, determined whether TFEB nuclear translocation and lysosomal activity could be regulated by treating HCT116 cells with the UQCRB inhibitor. Consistently, TFEB translocation into the nucleus was abolished after treatment of HCT116 cells with A1938 and mTP, indicating that TFEB activation was regulated by mROS production (Figure 6D,E). 

When HCT116 cells were treated with A1938 for 24 h under starvation conditions and stained with AO, there were significantly fewer acidic vesicles in A1938-treated HCT116 cells than in DMSO-treated HCT116 cells (Appendix A), indicating that A1938 prevented lysosome activity by inhibiting UQCRB activity (Figure 6F,G). Furthermore, the expression of the lysosomal protein LAMP1 was decreased after treatment with A1938 (Figure 6H). Taken together, these results indicate that inhibiting UQCRB protein reduced mROS levels, which decreased lysosomal activity.

## 3. Discussion

Mitochondria are important to the oxygen-sensing capabilities of cells [33]. UQCRB is especially an important oxygen sensor in hypoxia-induced [5] and VEGF-induced angiogenesis [34] plays a key role in this process. We previously reported that UQCRB overexpression affected mitochondrial morphology [8] and that mitochondrial dynamics were strongly associated with the clearance of damaged mitochondria [35]. Recent reports have also indicated that sustained elevation of ROS levels contributed to stress signaling in cells, which may damage mitochondria and lead to autophagy [4,35,36]. Although the associations between ROS and autophagy have been studied, these studies mainly used exogenous oxidants to increase mROS levels, which did not reflect a natural environment [35,36,37].

Although the connection between ROS and autophagy has been well studied [26], the role of UQCRB in autophagy is largely unknown. In the present study, we determined the role of UQCRB protein overexpression in autophagy, with respect to its effect on mROS production. Previously, Chang et al. generated a novel protein transduction domain (PTD)-conjugated wild type UQCRB fusion protein, and treatment with PTD-UQCRB generated mROS without cytotoxicity [14]. To further analyze autophagy, mutant expressing UQCRB and HCT116 cells were used in this study. The results are consistent with those of previous studies, which showed that the overexpression of UQCRB induced mROS generation, but this generation was reduced by treatment with the UQCRB inhibitor A1938 (Appendix A). We also showed that overexpression of UQCRB promoted autophagy flux, spontaneously induced Ca^2+^ release by TRPML1 lysosomal Ca^2+^ ion channels, and induced TFEB translocation (Figure 3). These results are consistent with those of previous studies showing that elevation in mROS levels led to TRPML1 activation, lysosomal Ca^2+^ release, and induction of TFEB-nuclear translocation [26].

TFEB promotes both autophagosome and lysosome biogenesis [27]. Lysosomal Ca^2+^ may also directly regulate lysosome biogenesis and autophagosome–lysosome fusion [26]. The present study showed that increased levels of mROS caused by UQCRB overexpression may be linked to lysosomal and autophagic pathways through TFEB. In mutant UQCRB-expressing cells, an increase in mROS levels led to TFEB-nuclear translocation and promoted lysosomal biogenesis (Figure 4D). Likewise, in HCT116 cells, lysosomal activity decreased after treatment with a UQCRB inhibitor, and lysosomal biogenesis was reduced by mROS regulation (Figure 6). Thus, the increase in mROS levels by the overexpression of UQCRB very likely could regulate the TFEB-lysosome biogenesis pathway [31,37].

ROS plays a significant role in autophagy by activating signaling pathways [38]. Tumor necrosis factor-α has been reported to induce autophagic cell death through a ROS-dependent mechanism [3]. Furthermore, many cancer cells have high levels of ROS and high frequencies of signaling events and mutations that increase ROS levels [3,39]. Mutations in mitochondrial DNA-encoded electron transport chain proteins have been reported in many types of human tumors [40]. For example, heteroplasmic mutations in complex I have been shown to increase mROS levels, colony formation rates in soft agar, and tumor formation in vivo [41]. These findings suggest that cancer cells show increased ROS production, which activates localized pro-tumorigenic signaling [3].

HCT116 CRC cells, which naturally overexpress UQCRB, were used to validate this finding. The overexpression of this protein resulted in the production of higher levels of mROS than control cells and subsequently affected autophagy (Figure 1 and Figure 6). Treating the cells with A1938, a UQCRB inhibitor, significantly reduced autophagy (Figure 6), and serum starvation led to cell death of HCT116 cells (Figure 1E). In addition, co-treatment of CQ and A1938 showed further reduced cell viability indicating that CQ and A1938 could play an additive role in autophagy inhibition. Under conditions that mimicked the physiology of humans, an increase in mROS levels may act as a survival signal to trigger autophagy to improve the quality of various physiological control mechanisms. However, the present study demonstrated that UQCRB activity reduced mROS production in HCT116 cells and that nutrient starvation made HCT116 cells more susceptible to A1938-induced cell death [20].

Moreover, HCT116 cells were xenografted into mice that were randomly administered either vehicle control or 10 mg/kg A1938. The tumor volumes of mice in the A1938 group decreased more than those in the vehicle group, whereas their body weights remained unchanged (Appendix A). In this in vivo study, we found that A1938 administration attenuated tumor growth. Thus, A1938 might be a good candidate for inducing tumor regression.

Although the role of autophagy in cancer cells remains controversial, we demonstrated that autophagy contributed to tumor cell survival in CRC [17,18]. Therefore, autophagy inhibitors are often included in chemotherapeutic drug regimens for the treatment of cancers [42]. The autophagic flux of HCT116 cells significantly increased in response to nutrient stress, and autophagy inhibition was more sensitive in vivo than in normal colon cells [17]. These findings demonstrated that blocking autophagy decreased tumor growth (Appendix A). UQCRB variants play important roles in various cancers, such as hepatocellular carcinoma, ovarian cancer, and pancreatic ductal adenocarcinoma [43,44,45,46]. It would therefore be important to investigate how UQCRB is related to the pathophysiological roles of autophagy during tumorigenesis.

## 4. Materials and Methods

### 4.1. Compounds and Antibodies 

A1938 was synthesized by our group [11]. DMSO (D2650), bafilomycin A1 (B1793), CQ (C6628), rapamycin (553210), mito-TEMPO, AO (A6014), and Triton X-100 were purchased from Sigma-Aldrich (St. Louis, MO, USA). LysoTracker Deep Red (L12492), Hoechst33342 (H3570), Lipofectamine 2000 (52887), Plus Reagent (10964), protease, phosphatase inhibitor solution (78441), Dulbecco’s Modified Eagle’s Medium (DMEM), Roswell Park Memorial Institute Medium (RPMI), fetal bovine serum (FBS), and antibiotics were purchased from Invitrogen ThermoFisher Scientific (Grand Island, NY, USA). Anti-UQCRB was obtained from Sigma-Aldrich; anti-LC3B, anti-actin, and anti-LAMP1 were obtained from Abcam (Cambridge, UK). Anti-p62 was obtained from BD Biosciences (Franklin Lakes, NJ, USA), and anti-TFEB and anti-GAPDH were from Cell Signaling Technology (Danvers, MA, USA). Anti-Lamin A/C was obtained from Santa Cruz Biotechnology (Dallas, TX, USA). The mRFP-GFP-LC3B plasmids were kindly provided by Dr. Jaewhan Song of Yonsei University (Seoul, Korea).

### 4.2. Cell Culture

Cell lines were purchased commercially by American Type Culture Collection (ATCC), CCD18co (CCL-1459TM, ATCC), HCT116 (CCL-247 TM, ATCC), HEK293 (CRL-1573.3 TM, ATCC). The mutant UQCRB-overexpressing cell line (MT) was established previously by our group and deposited at NCBI (ID-SAMN13680025) [10]. HEK293, MT, and CCD18co cells were grown in DMEM, supplemented with 10% FBS and 1% antibiotics. HCT116 cells were grown in RPMI1640 with the same conditions as described above. All cells were incubated at 37 °C in a humidified incubator with 5% CO_2_ at a pH of 7.4. To maintain the stability of the mutant cell lines, 1 mg/mL G418 was added to the medium on a regular basis.

### 4.3. Immunoblotting

Soluble proteins were harvested from cells using a SDS lysis buffer comprising 50 mM Tris HCl with a pH of 6.8, containing 10% glycerol, 2% SDS, 10 mM dithiothreitol, and 0.005% bromophenol blue. Equal concentrations of proteins were separated by 11% or 12.5% SDS-PAGE and transferred to polyvinylidene fluoride membranes (EMD Millipore, Billerica, MA, USA). Blots were then blocked and immunolabeled overnight at 4 °C with the following primary antibodies: anti-actin (Abcam, ab6276), LAMP1 (Abcam, ab24170), anti-LC3B (Cell Signaling Technology, 2775), anti-Myc (Abcam, ab18185), anti-p62 (BD Biosciences), anti-UQCRB (Abcam, ab190360). Immunolabeling was visualized using an enhanced chemiluminescence kit (Amersham Life Science, Chalfont, UK), according to the manufacturer’s instructions. Images were quantified using Image Lab software (Bio-Rad, Hercules, CA, USA). Actin and GAPDH were used as internal controls. All band intensities were proportional to the amount of target protein on the membrane as determined by the linear range of detection. Images were quantified using Image LabTM software (Bio-Rad).

### 4.4. Immunocytochemistry

To determine changes in autophagy, both WT HEK293- and mutant UQCRB-expressing cell lines were seeded at a density of 1.5 × 10^5^ cells/well in 6-well plates and incubated overnight. The medium was then changed to serum-free DMEM (Gibco, Rockville, MD, USA) and incubated for 24 h. After starvation, the cell samples were fixed with 4% formaldehyde (Sigma-Aldrich), diluted in phosphate-buffered saline (PBS), and washed with 1× PBS three times. The cells were then treated with LC3 primary antibody (Abcam, ab48394) for 2 h. Nuclei were stained with Hoechst (Invitrogen, Grand Island, NY, USA). Images were obtained using an LSM980 confocal microscope at 400× magnification.

### 4.5. mROS Measurements

Mitochondria ROS levels were assessed using the MitoSOX red fluorescence mitochondrial superoxide indicator (Invitrogen). The cells were incubated with 5 μM MitoSOX and Hoechst33342 (Life Technologies, Grand Island, NY, USA) for 10 min, washed with medium or 1× PBS three times, and fixed with 4% formaldehyde. MitoSOX and Hoechst staining results were analyzed using a Zeiss LSM 980 confocal microscope (Carl Zeiss MicroImaging, Thornwood, NY, USA), and the fluorescence intensity of MitoSOX was measured using ImageJ software (National Institutes of Health, Bethesda, MD, USA).

### 4.6. The mRFP-GFP-LC3B Plasmid Transfection

Cells were transfected with the mRFP-GFP-LC3B plasmid using Lipofectamine 2000 transfection reagent (Invitrogen) for 24 h. The cells were then treated with DMSO control, A1938, or mitoTEMPO for 24 h. Nuclei were stained with Hoechst33342. Following incubation for 10 min, the cells were fixed with 4% formaldehyde and washed three times with medium or PBS. Images were obtained using an LSM700 confocal microscope at 400× magnification.

### 4.7. GCaMP3-ML1 Ca^2+^ Imaging

Cells were grown on 15 mm coverslips and transfected with a plasmid encoding a lysosomal GCaMP3-ML1 Ca^2+^ probe. The experiment was conducted 3−5 h after plating while cells still exhibited a round morphology. Lysosomal Ca^2+^ release was measured in a basal Ca^2+^ solution containing 145 mM NaCl, 5 mM KCl, 3 mM MgCl_2_, 10 mM glucose, 1 mM EGTA, and 20 mM HEPES (pH 7.4) with or without 40 μM ML-SA1 and 1 μM ionomycin by monitoring the fluorescence intensity at 470 nm with an LSM980 confocal microscope (Zeiss, Wetzlar, Germany).

### 4.8. EGFP-TFEB Nuclear Translocation Assay

HEK293 and MT cells were seeded into 6-well plates and incubated overnight. The enhanced green fluorescence protein-TFEB vector was then transfected into the cells using Lipofectamine 2000 reagent (Invitrogen), according to the manufacturer’s instructions. Then, A1938 treatment was conducted for 6 h as indicated. Cells were then fixed with 4% formaldehyde and washed three times with medium or PBS. Nuclei were stained with Hoechst33342. Images were obtained using a LSM700 confocal microscope at 400× magnification.

### 4.9. Fluorescence Staining

HEK293 and MT cells were grown on 15 mm coverslips at a density of 1.3 × 10^5^ cells/well in 6-well plates for AO staining. The cells were then treated with drugs for 24 h and then treated with 5 μg/mL AO (Sigma-Aldrich). Nuclei were stained with Hoechst. The cells were then incubated for 20 min, fixed with 4% paraformaldehyde (PFA), and washed three times with PBS. Images were obtained using an LSM880 confocal microscope at 400× magnification. Red fluorescence intensity was quantified using ImageJ software. For LysoTracker Deep Red (L12492) staining, HEK293 and MT cells were grown on 15 mm coverslips at a density of 1.3 × 10^5^ cells/well in 6-well plates. The cells were then treated with drugs for 24 h, then treated with 100 nM LysoTracker Deep Red for 20 min, fixed with 4% PFA, and washed three times with medium or PBS. Images were obtained using an LSM700 confocal microscope at 400× magnification. The number of cell red puncta was quantified using ImageJ software.

### 4.10. Cell Proliferation Assay

Cell growth was measured using the MTT colorimetric assay. HCT116 cancer cells were seeded at a density of 5 × 10^3^ cells/well in 96-well plates and incubated overnight. Cells were then treated with A1938 for various concentrations and durations. At the end of the assay, 2 mg/mL MTT was added to each well and incubated for 4 h. MTT-formazan was dissolved in each well in 150 μL DMSO, and the absorbance was read at 540 nm with a microplate reader (Bio-Tek Instruments, Winooski, VT, USA).

### 4.11. Cell Death Assay

HCT116 cells were seeded at 2 × 10^4^ cells/well in 24-well plates and incubated overnight. The medium was then changed to serum-free DMEM (Gibco), and an inhibitor of cell death (CQ) was added. After 72 h, cell viability was determined using trypan blue staining.

### 4.12. Statistics

All data are expressed as the mean ± SD or mean ± SEM, as determined using Prism software, version 9.00 for Windows (GraphPad Software, San Diego, CA, USA). Statistical analyses were performed using an unpaired, two-tailed Student’s *t*-test. A value of *p* < 0.05 was considered statistically significant. 

Appendix A can be found at Appendix A.

## 5. Conclusions

This study showed that UQCRB overexpression-induced autophagy may occur via generated mROS and that UQCRB is a potential therapeutic target for autophagy regulation in UQCRB-related diseases, such as CRC.

This study determined the UQCRB protein expression and mROS generation levels of mutant UQCRB-expressing cell lines and HCT116 cells. UQCRB overexpression promoted autophagy flux by lysosomal Ca^2+^ release through the TRPML1 channel, which led to TFEB translocation into the nucleus to promote both autophagosome and lysosome biogenesis. Additionally, autophagy and lysosomal activity induced by UQCRB overexpression were abolished by the UQCRB inhibitor, A1938, in MT and HCT116 cells. A1938-induced HCT116 cell death occurred in serum-free conditions that mimicked nutrient-deprived environments. In addition, the HCT116 xenograft study showed that the A1938 treatment decreased tumor volume but did not affect the body weights of mice. 

Although A1938 has been shown to have a therapeutic value to treat cancer cells particularly overexpressing UQCRB protein, high micromolar levels of A1938 were needed in our previous studies [9,15]. To overcome these limitations, it is necessary to improve the pharmacological properties of A1938 by designing and synthesizing a series of A1938 analogs.

Collectively, the results demonstrated that UQCRB, a crucial mediator of mROS generation, was a potential therapeutic target for autophagy regulation. Small molecules that specifically regulate the function of UQCRB could suppress tumor growth by regulating the functional links between UQCRB, mROS, and autophagy. Therefore, UQCRB inhibitors could be a novel therapeutic strategy for regulating autophagy in the tumor environment.

## Figures and Tables

**Figure 1 cancers-14-01883-f001:**
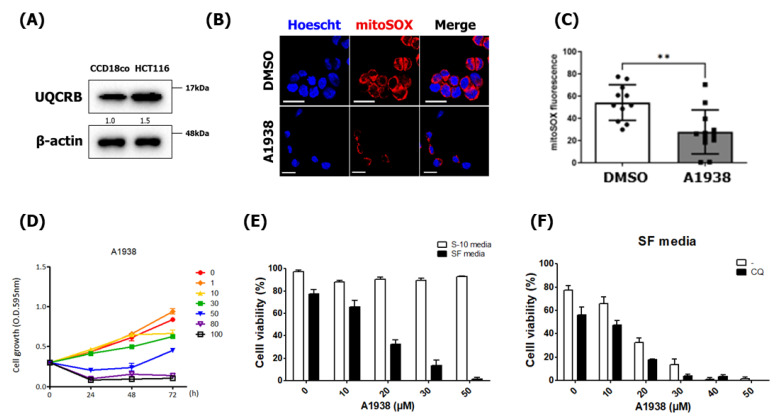
A1938-induced cell death is sensitized in HCT116 cells. (**A**) Expression levels of UQCRB in CCD18co normal human colon cells and HCT116 CRC cells. β-actin was used as an internal control. The relative band intensities value was normalized to β-actin (loading control). The normalized protein levels are shown under each band. (**B**) Confocal microscopy images of MitoSOX Red fluorescence showing mROS generated in HCT116 cells. (**C**) The mean ± SD of cells (*n* = 11) treated with 30 µM A1938 for 4 h. Scale bar, 20 μm. (**D**) HCT116 cell viability after treatment with 0, 1, 10, 30, 50, 80, or 100 μM A1938 as assessed using the MTT assay. (**E**) The viability of A1938-treated HCT116 cells using concentrations of 0, 10, 20, 30, or 50 μM in the trypan blue assay. S-10 media means medium containing 10% serum, and SF media does medium without the serum. SF media have been used for establishing the serum-starvation conditions. (**F**) HCT116 cell viability as measured using ImageJ software after treatment with 0−50 μM A1938 in the presence or absence of 5 μM chloroquine for 3 days in a serum-free medium. Statistical significance was assessed using the Student’s *t*-test. ** *p* < 0.01.

**Figure 2 cancers-14-01883-f002:**
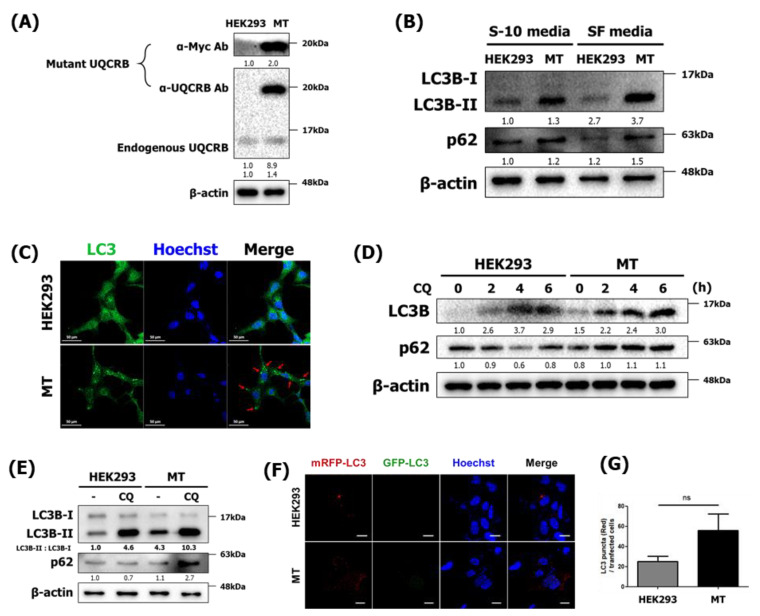
Autophagy is induced in MT cells. (**A**) Mutant UQCRB-expressing cell lines were examined by Western blotting using anti-UQCRB and anti-Myc antibodies. (**B**) HEK293 and MT cells treated with either 10% serum-containing medium or serum-free medium for 24 h and then analyzed by Western blot analysis using antibodies against LC3B and p62. Appendix A shows treatment of either 10% serum-containing medium or serum-free medium for 4 h. (**C**) Confocal microscopy images of HEK293 and MT cells treated with serum-free medium for 24 h after immunostaining with anti-LC3 antibody. Scale bar, 50 μm. (**D**) Western blot analysis results of LC3B and p62 levels in cells after treatment with 5 μM chloroquine (CQ) treatment for 0–6 h. (**E**) HEK293 and MT cells were treated with either deionized water (DIW) or CQ for 6 h. Cell extracts were subjected to Western blotting using LC3B and p62 antibodies. The densitometric value of LC3-II was quantitated based on Western blot data and normalized to LC3-I levels. (**F**) Confocal microscopy images of HEK293 and MT cells transfected with mRFP-GFP-LC3 for 24 h. Scale bar, 20 μm. (**G**) Numbers of mRFP-LC3 puncta were counted in HEK293 and MT cells (*n* = 8). The relative band intensities value was normalized to β-actin (loading control). The normalized protein levels are shown under each band. Data are presented as the mean ± SD; statistical significance was assessed using an unpaired *t*-test. ns, *p* > 0.05.

**Figure 3 cancers-14-01883-f003:**
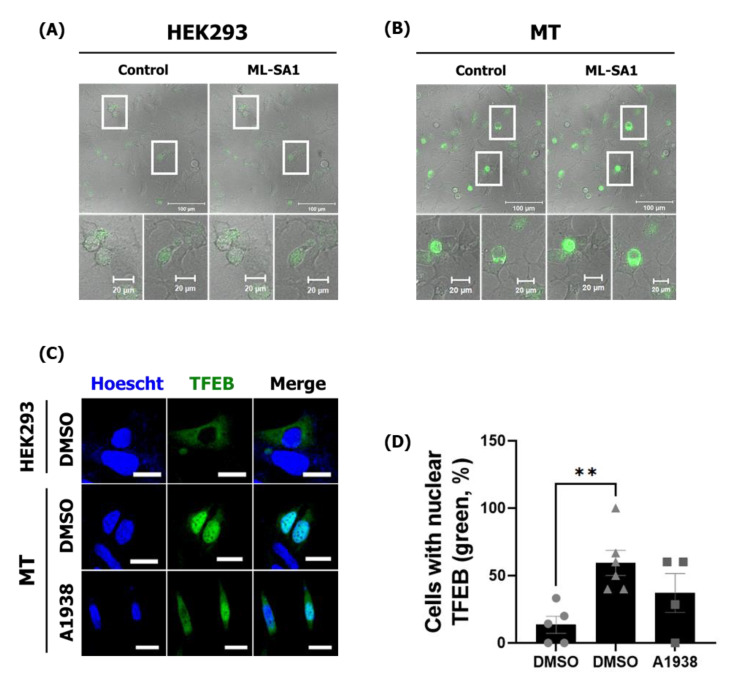
TRPML1-induced lysosomal Ca^2+^ release and TFEB activation are increased in MT cells. HEK293 and MT cells were transfected with GCaMP3-TRPML1 encoding a lysosome-specific Ca^2+^ probe, 40 µM ML-SA1, for 120 s. (**A**) Confocal microscopy images of GCaMP green fluorescence showing HEK293 and (**B**) MT cells. ML-SA1 treatment for 120 s. (**C**) Confocal microscopy images of HEK293 and MT cells transfected with EGFP-TFEB and treated with the DMSO control or A1938 for 6 h. (**D**) The number of cells with a nuclear (Nuc) or cytoplasmic (Cyt) TFEB localization ratio is represented in the graph. Scale bar, 20 μm. Statistical significance was assessed using Student’s *t*-test. ** *p* < 0.01.

**Figure 4 cancers-14-01883-f004:**
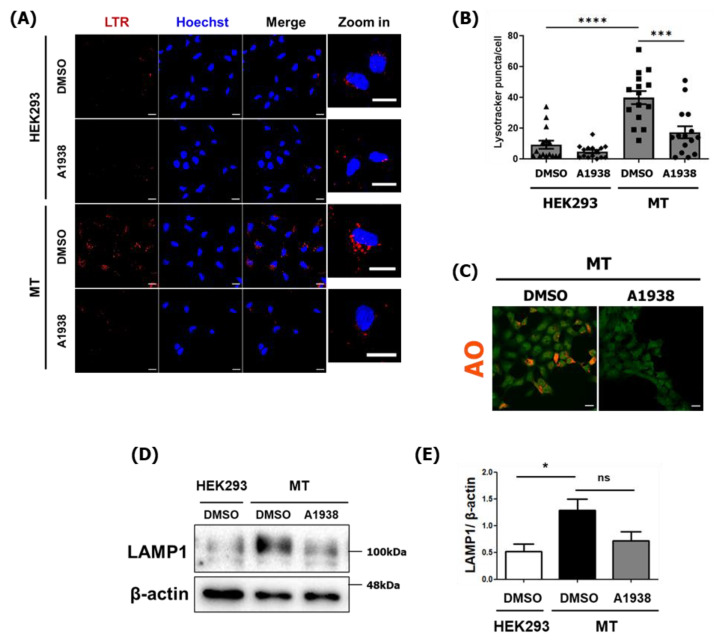
Lysosome activity in MT cells with or without A1938 treatment. (**A**) Confocal microscopy images and (**B**) numbers of acidic lysosome red puncta were counted in HEK293 and MT cells (*n* = 15). The cells were treated with DMSO as a control or 30 µM A1938 for 24 h and then stained with LTR. Scale bar, 20 μm. (**C**) Confocal microscopy images of MT cells treated with DMSO as a control or 30 µM A1938 for 24 h, then the live cells were stained with 2 μg/mL acridine orange for 25 min and fixed. Scale bar, 20 μm. (**D**) Images of HEK293 and MT cell extracts treated with the DMSO control or 30 µM A1938 for 24 h in a serum-free medium, then subjected to Western blot analysis using antibodies against LAMP1. (**E**) Intensities of the LAMP1 immunoblot bands were normalized to β-actin expression. Western blot analysis out of three independent experiments. Statistical significance was assessed using Student’s *t*-test. ns, *p* > 0.05, * *p* < 0.05, *** *p* < 0.001, **** *p* < 0.0001.

**Figure 5 cancers-14-01883-f005:**
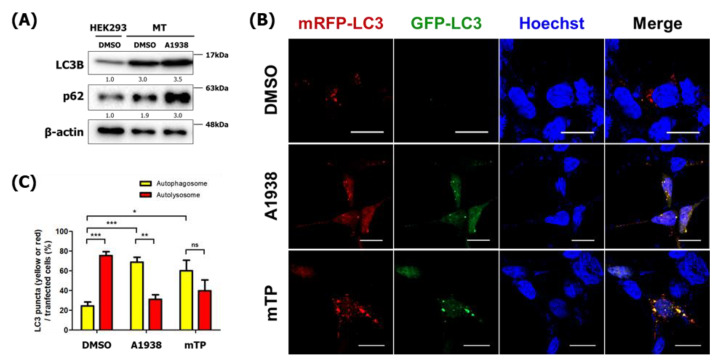
A1938 regulates autophagy flux by suppression of mROS generation in MT cells. (**A**) Cells treated with A1938 for 24 h in a serum-free medium. The relative band intensities value was normalized to β-actin (loading control). The normalized protein levels are shown under each band. (**B**) Confocal microscopy images of HEK293 and MT cells transfected with mRFP-GFP-LC3 and treated with DMSO, A1938, or Mito-Tempo (mTP) for 24 h. Scale bar, 20 μm. Quantification of data indicating red puncta (autolysosome) versus yellow puncta (autophagosome) is shown in (**C**). Values are the means ± SEM; statistical significance was assessed using a paired *t*-test. *n* = 10 cells, ns, *p* > 0.05, * *p* < 0.05, ** *p* < 0.01, *** *p* < 0.001.

**Figure 6 cancers-14-01883-f006:**
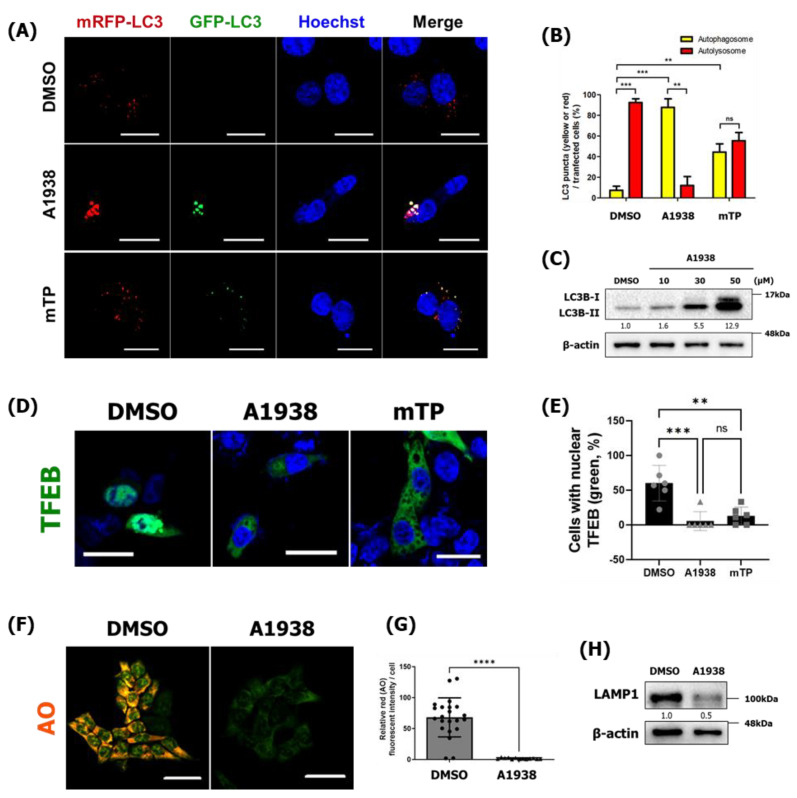
A1938 impairs autophagy through the downregulation of mROS in HCT116 cells (**A**) Confocal microscopy image of HCT116 cells transfected with mRFP-GFP-LC3 and treated with DMSO, A1938, or mTP for 24 h. Scale bar, 20 μm. Quantification of data indicating red puncta (autolysosome) versus yellow puncta (autophagosome) is shown in (**B**). Values are the means ± SEM; statistical significance was assessed using a paired *t*-test. *n* = 10 cells, ns, *p* > 0.05, ** *p* < 0.01, *** *p* < 0.001. (**C**) HCT 116 cells treated with various concentrations of A1938 for 24 h in a serum-free medium. The resulting cell extracts were subjected to Western blot analysis using antibodies against LC3B. (**D**) Confocal microscopy images of HCT116 cells transfected with GFP-TFEB and treated with the DMSO control or A1938 for 6 h. (**E**) The number of cells with a nuclear (Nuc) or cytoplasmic (Cyt) TFEB localization ratio is represented in the graph. (**F**) HCT116 cells were treated with DMSO control or 30 µM A1938 for 24 h. Live cells were stained with 2 μg/mL acridine orange (AO) for 25 min, fixed, and examined by confocal fluorescence microscopy. Confocal fluorescence microscopy images of HCT116 cells treated with A1938 in RPMI medium containing 10% serum. (**G**) The average AO intensity per HCT116 cell treated with A1938 in RPMI medium containing 10% serum (*n* > 10). (**H**) HCT116 cells treated with 30 µM A1938 for 24 h and then subjected to Western blot analysis using antibodies against LAMP1. Scale bar, 20 μm. Statistical significance was assessed using Student’s *t*-test. ns, *p* > 0.05, ** *p* < 0.01, *** *p* < 0.001, **** *p* < 0.0001. The relative band intensities value was normalized to β-actin (loading control). The normalized protein levels are shown under each band. All original western blots can be found at Appendix A.

## Data Availability

Data are contained within the article.

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
