# Peer review of "Mitochondrial ROS Produced in Human Colon Carcinoma Associated with Cell Survival via Autophagy"

_cancers, 2022, doi:10.3390/cancers14081883_

Round 1

Reviewer 1 Report

The authors did not adequately address two of my concerns.

In response to my comment:  "In Figure 5C, the authors should specify what was compared in the statistical analysis, so that the reader can know if the mTP was significantly different from the DMSO, the A1938 or both." they described the data that they had used in the graph, but they did not address the statistical analysis. They should indicate which two groups are compared for the "****" or "ns" shown on the graph.

In response to my comment: "The grammar of the following two sentences starting on Line 130 should be corrected: “Increased autophagy activation in HCT116 cells involving recycling of cell components to facilitate survival in a nutrient deprived environment” and “However, inhibition of autophagy flux in HCT116 cells in serum starvation condition would make these cell could not afford the nutrient by recycling of cell components.”

The revised second sentence also does not make sense.  Here is a suggestion/example of how this sentence could be changed to increase clarity: "However, inhibition of autophagic flux in HCT116 cells under serum starvation conditions would prevent autophagic recycling of cell components to allow survival.

Reviewer 2 Report

The manuscript entitled as “Mitochondrial ROS Produced in Human Colon Carcinoma HCT116 Cells Associated with Cell Survival via Autophagy” have demonstrated interesting results. However, there is definitely a chance of improvement in terms of technical and experimental details. Therefore, I recommend this authors to put light on the below mentioned comments:

1. Why authors have mentioned a particular cancer cell line in the title? I recommend to change the title appropriately.

2. In order to comment on the role of UQCRB in CRC, authors must use at least three UQCRB wild type expressing colon cancer cells. Using only one cell line is not sufficient enough to generalize the function of UQCRB in CRC.

3. In order to investigate that absence of UQCRB protein decreases mROS or the partial inhibition of UQCRB is efficient enough to regulate the mROS, UQCRB should be knocked out. Because, small molecules inhibitors are not always limited to one specific target proteins. Therefore, UQCRB should be genetically silenced.

4. I strongly suggest authors to overexpress  UQCRB in HCT116 cell line, in order to keep the same genetic background expect the manipulation in UQCRB expression. Or else if it is just about the mechanism of study based on its genetic background they should not specify any organs and cancer type.

5. Why authors have suddenly changed the cell line? It should be discussed in the manuscript, if there is any rational behind it.

6. Authors must show the results of A1938 treatment against HEK293 cell viability.

Round 2

Reviewer 2 Report

I think authors have made appropriate changes where ever it was necessary keeps the comments in mind. I believe, at the current formate the manuscript can be accepted.

This manuscript is a resubmission of an earlier submission. The following is a list of the peer review reports and author responses from that submission.

Round 1

Reviewer 1 Report

Eunji Gwak et al.’s manuscript “Mitochondrial ROS Produced in Human Colon Carcinoma HCT116 Cells Reduces Cell Survival via Autophagy’’ investigated the effect of Mitochondrial ROS (mROS), UQCRB on cell death/Cell Survival via modulation of autophagy. Generally, this study needs to pay more attention to the basic principle of autophagy research methods and present results logically.

  • Please refer to the Guidelines for the use and interpretation of assays for monitoring autophagy (4th edition) https://www.tandfonline.com/doi/full/10.1080/15548627.2020.1797280. Especially how to present and design the experiment to study the autophagic flux.
  • All western blot results in all figures need to show the two bands (I and II) of LC3B, as well as the size of each antibody. Moreover, the bar char to indicate the densitometric value is also needed.
  • The first sentence in the Abstract, “Human colon carcinomas, including HCT116 cells, often exhibit high basal autophagic flux under nutrient deprivation or hypoxic conditions’ is strange to me; if there is nutrient deprivation, then this autophagic flux cannot take as the basal level.
  • ‘In addition, UQCRB is more frequently highly expressed in CRC patients whose diseases are more advanced and who have poor pathological differentiation. The expression level of UQCRB-related circulating miR-4435, which regulates the invasion of HCT116 cells and migration functions, is upregulated in HCT116’. Can the author explain why the UQCRB and miR-4435 have the same trend in the high expression? To me, higher expression of miRNA level usually depresses gene expression.
  • For the starvation treatment, 24 hours is too long to study autophagy. Can the author demonstrate the short time starvation does not work in your system?
  • Where are the results for CQ treatment in HCT16? Figure 1D is for A1938, but not CQ results.
  • ‘These results suggested that targeting UQCRB activity by A1938, a UQCRB inhibitor, could inhibit autophagy in CRC cells’. However, this conclusion is not solid. The authors should show clear evidence of how the A1938 inhibits autophagy. For example, the wb results for LC3B, p62, and GFP-puncta which are all missing in Figure 1.
  • Section 2.2, UQCRB MT promotes the mROS level, and MT had higher LC3B and p62 levels. The results cannot support the relationship between mROS and autophagy induction in your system. Moreover, lacking autophagy usually leads to accumulation of p62 but not decrease. On the other hand, autophagy induction will reduce the p62 level.
  • All the GFP-LC3 or RFP-LC3 figures should clearly show the LC3 puncta.
  • Can the author exclude those green dots that are not dead cells?
  • The quality of Figure 4D is poor. Please improve it and add the statistic result in Figure 4 E.
  • Section 2.5.” MT cells treated with A1938 exhibited in-creased LC3B-II and p62 protein levels’ however, the conclusion in this part is “….which indicates that A1938 decreased autophagy flux.’ Can you explain why? Moreover, A1938 increases LC3B-II in figure 5A but decease LC3B-II in figure 5D in the same MT cells. Why?
  • In figure 6C, the author again showed that A1938 dramatically increased LC3B-II. As we know, salvation alone also can induce autophagic flux. The author needs to add control to get a solid conclusion.

Author Response

February 9, 2022

Dear Prof. Dr. Samuel C. Mok, Editor-in-Chief, Cancers

CC: Prof. Dr. Gyorgy Marko-Varga, Guest editor, Cancers

Ms. Lillian Cao, Assistant Editor, Cancers

We are grateful to reviewers for their time and valuable comments on our manuscript entitled “Mitochondrial ROS Produced in Human Colon Carcinoma HCT116 Cells Reduces Cell Survival via Autophagy” (cancers-1585152). We have fully addressed reviewers’ concerns and extensively revised the manuscript accordingly. Followings are point-to-point responses to comments by the reviewers.

  • Revised portions were highlighted in red in the revised manuscript.

Reviewers' comments:

Reviewer #1:

Eunji Gwak et al.’s manuscript “Mitochondrial ROS Produced in Human Colon Carcinoma HCT116 Cells Reduces Cell Survival via Autophagy’’ investigated the effect of Mitochondrial ROS (mROS), UQCRB on cell death/Cell Survival via modulation of autophagy. Generally, this study needs to pay more attention to the basic principle of autophagy research methods and present results logically.

#1. Please refer to the Guidelines for the use and interpretation of assays for monitoring autophagy (4th edition) https://www.tandfonline.com/doi/full/10.1080/15548627.2020.

  1. Especially how to present and design the experiment to study the autophagic flux.

Response: We appreciate the Reviewer’s suggestion. In this guideline, authors describe that the relevant parameter in LC3 assays is the difference in the amount of LC3-II in both the presence and absence of saturating levels of inhibitors, which can be used to examine the transit of LC3-II through the autophagic pathway; if flux is occurring, the amount of LC3-II will be higher in the presence of the inhibitor. Lysosomal degradation can be prevented through the use of protease inhibitors (e.g., pepstatin A, leupeptin and E-64d), compounds that neutralize the lysosomal pH such as bafilomycin A1, CQ or NH4Cl, or by treatment with agents that block the fusion of autophagosomes with lysosomes (note that CQ blocks autophagy predominantly by inhibiting autophagosome-lysosome fusion and that bafilomycin A1 will ultimately cause a fusion block as well as neutralize the pH, but the inhibition of fusion may be due to a block in ATP2A/SERCA activity) [1]. Therefore, we designed autophagy flux experiment using CQ that neutralizes lysosomal pH and inhibits lysosomal fusion in Figure 2D-E. The densitometric intensity ratio of LC3-II:LC3-I indicated that MT cells exhibited a higher LC3-II conversion rate than that of control cells with fully accumulated LC3-II and P62 by CQ treatment. To further validate the effect of UQCRB on autophagy flux, a double-tagged mRFP-GFP-LC3 plasmid, which is also indicated in the guideline as ‘a fluorescence assay that is designed to monitor flux’, was used to visualize the transition from neutral autophagosomes to acidic autolysosomes. Red puncta fluorescence was increased in MT cells compare to normal HEK293 cells (Figure 1F), decreased upon A1938 or mTP treatment in MT cells (Figure 5B-C). These experiment were designed to study the autophagic flux and we added this interpretation in the revised manuscript.

<Reference>

  1. Klionsky, D.J.; Abdel-Aziz, A.K.; Abdelfatah, S.; Abdellatif, M.; Abdoli, A.; Abel, S.; Abeliovich, H.; Abildgaard, M.H.; Abudu, Y.P.; Acevedo-Arozena, A. Guidelines for the use and interpretation of assays for monitoring autophagy. autophagy 2021, 17, 1-382.

#2. All western blot results in all figures need to show the two bands (I and II) of LC3B, as well as the size of each antibody. Moreover, the bar char to indicate the densitometric value is also needed.

Response: Thank you for your valuable points. It is our regret, however, that LC3B-I bands were hardly detected in our experiments so it was inevitable to show only the one bands of LC3B (Figure 2D). Some LC3 blot images did not show LC3-I bands due to exposure time. We replaced long exposure time images displaying LC3-I band size respectively (Figure 2B, Figure 6C). However, it is hard to compare the LC3B-II band in Figure 5A’s long exposure time image (as seen below Supplementary Data) so we would like to keep previous data of the short exposure time image. In addition, we could check some studies conducted from other groups also produced only one band of LC3 in the western blot results [2,3]. In Figure 5D and 5E from the Marina N. et al.’s study, they marked LC3B-I site to the western blot image but the band barely shown in the image [2]. In Liu C. et al’s study, they have been cropped LC3-II band and did not show either the original whole blot image or LC3B-I band [3]. In our study, the forms of the LC3 protein were detected using anti-LC3 antibodies (Cell Signaling Technology, 2775) and by the supporting data, expecting molecular weight for LC3B-I is 16 kDa and LC3B-II is 14 kDa. All data shown in figures, the one band of LC3B was observed between 17 kDa and 11kDa, however, it was much closer to the 11 kDa ladder band. Although some of western blot results are lack of LC3B-I band, however, according to the Guidelines for the use and interpretation of assays for monitoring autophagy (4th edition), levels of LC3-II should be compared not to LC3-I, but ideally to more than one housekeeping protein such as ACTB/β-actin [1]. Based on this guideline, we try to interpret the pattern of LC3 protein by the densitometric value of LC3B-II and β-actin. However, we will try to get better LC3B two bands images in coming investigations.

In addition, densitometric value and size of ladder protein near each antibody of the western blot results were added into Figure 1A, Figure 2A, Figure 2B, Figure 2D, Figure 2E, Figure 5A, Figure 6C and Figure 6H accordingly.

Supplementary Data. Figure 5A long exposure time image.

<Reference>

  1. Klionsky, D.J.; Abdel-Aziz, A.K.; Abdelfatah, S.; Abdellatif, M.; Abdoli, A.; Abel, S.; Abeliovich, H.; Abildgaard, M.H.; Abudu, Y.P.; Acevedo-Arozena, A. Guidelines for the use and interpretation of assays for monitoring autophagy. Autophagy 2021, 17, 1-382.
  2. Sharifi, M.N.; Mowers, E.E.; Drake, L.E.; Collier, C.; Chen, H.; Zamora, M.; Mui, S.; Macleod, K.F. Autophagy promotes focal adhesion disassembly and cell motility of metastatic tumor cells through the direct interaction of paxillin with LC3. Cell reports 2016, 15, 1660-1672.
  3. Liu, C.; Fu, H.; Liu, X.; Lei, Q.; Zhang, Y.; She, X.; Liu, Q.; Liu, Q.; Sun, Y.; Li, G. LINC00470 coordinates the epigenetic regulation of ELFN2 to distract GBM cell autophagy. Molecular therapy 2018, 26, 2267-2281.

#3. The first sentence in the Abstract, “Human colon carcinomas, including HCT116 cells, often exhibit high basal autophagic flux under nutrient deprivation or hypoxic conditions’ is strange to me; if there is nutrient deprivation, then this autophagic flux cannot take as the basal level.

Response: Thank you for point out the error. We agree with the reviewer’s points that using the word ‘high’ in this context is not appropriate. Therefore, we exclude this word in sentence throughout the revised manuscript.

#4. ‘In addition, UQCRB is more frequently highly expressed in CRC patients whose diseases are more advanced and who have poor pathological differentiation. The expression level of UQCRB-related circulating miR-4435, which regulates the invasion of HCT116 cells and migration functions, is upregulated in HCT116’. Can the author explain why the UQCRB and miR-4435 have the same trend in the high expression? To me, higher expression of miRNA level usually depresses gene expression.

Response: As the reviewer pointed out, miR4435 is not a micro RNA targeting the UQCRB’s mRNA. In our previous studies, we have identified miR4435 as UQCRB-related key miRNA targeting a tumor suppressor gene TIMP3 [4]. By targeting TIMP3 mRNAs, thus suppresses the expression level of TIMP3. In addition, the expression of miR-4435 was suppressed by UQCRB inhibitor treatment whereas TIMP3 was up-regulated. TIMP3 was also upregulated in response to miR-4435 inhibitor and UQCRB inhibitor treatments.

<Reference>

  1. Hong, J.W.; Kim, J.M.; Kim, J.E.; Cho, H.; Kim, D.; Kim, W.; Oh, J.-W.; Kwon, H.J. MiR-4435 is an UQCRB-related circulating miRNA in human colorectal cancer. Scientific reports 2020, 10, 1-11.

#5. For the starvation treatment, 24 hours is too long to study autophagy. Can the author demonstrate the short time starvation does not work in your system?

Response: We appreciate for very valuable suggestions. In amino acid starvation condition, many reference articles indicate short time starvation (around 1~4 h) for the cascade of mTOR inhibition-TFEB-autophagy [5]. In our experiment, however, we designed another starvation condition, serum deprivation, a mimic of tumor environment. In the condition, we tried short time starvation (4h) before, but did not show any response in autophagy induction (supplementary Figure 4.). Moreover, many references portray long term starvation (24 h) for autophagy induction [6,7]. Therefore, we thought that 24 h of starvation with drug treatment is reliable for our system.

<Reference>

  1. Napolitano, G.; Esposito, A.; Choi, H.; Matarese, M.; Benedetti, V.; Di Malta, C.; Monfregola, J.; Medina, D.L.; Lippincott-Schwartz, J.; Ballabio, A. mTOR-dependent phosphorylation controls TFEB nuclear export. Nature communications 2018, 9, 1-10.
  2. Belleudi, F.; Nanni, M.; Raffa, S.; Torrisi, M.R. HPV16 E5 deregulates the autophagic process in human keratinocytes. Oncotarget 2015, 6, 9370.
  3. Chen, P.-S.; Wang, K.-C.; Chao, T.-H.; Chung, H.-C.; Tseng, S.-Y.; Luo, C.-Y.; Shi, G.-Y.; Wu, H.-L.; Li, Y.-H. Recombinant thrombomodulin exerts anti-autophagic action in endothelial cells and provides anti-atherosclerosis effect in apolipoprotein E deficient mice. Scientific reports 2017, 7, 1-9.

#6. Where are the results for CQ treatment in HCT116? Figure 1D is for A1938, but not CQ results.

Response: Thank you for valuable comments. In this study, we used CQ as a control drug as an autophagy inhibitor so we did not show MTT assay of CQ in HCT116 cells. However, several studies already showed CQ treatment result in HCT116 cells [8]. For instance, Schonewolf C.A. et al. showed MTT assay in HCT116 cells 72 h after treatment with 10 μM CQ treatment and cell viability of HCT116 cells was significantly decreased [8].

<Reference>

  1. Schonewolf, C.A.; Mehta, M.; Schiff, D.; Wu, H.; Haffty, B.G.; Karantza, V.; Jabbour, S.K. Autophagy inhibition by chloroquine sensitizes HT-29 colorectal cancer cells to concurrent chemoradiation. World journal of gastrointestinal oncology 2014, 6, 74.

#7. ‘These results suggested that targeting UQCRB activity by A1938, a UQCRB inhibitor, could inhibit autophagy in CRC cells’. However, this conclusion is not solid. The authors should show clear evidence of how the A1938 inhibits autophagy. For example, the wb results for LC3B, p62, and GFP-puncta which are all missing in Figure 1.

Response: We are grateful to the reviewer to point out the important issue that we did not fully provide an explanation behind the result. Since LC3B, p62, and double-tagged mRFP-GFP-LC3 plasmid assay was conducted in Figure 6, we removed these parts from the current manuscript and focused on Figure 1’s result. As we agree with the reviewer’s point that former conclusion is not solid, therefore manuscript and Figures were rearranged accordingly. Described below is our revised interpretation of the observation:

à Increased autophagy activation in HCT116 cells involving recycling of cell components to facilitate survival in a nutrient deprived environment. However, inhibition of autophagy flux in HCT116 cells in serum starvation condition would make these cell could not afford the nutrient by recycling of cell components. Taken together, these results suggested that inhibiting autophagy flux in HCT116 cells led to increased cytotoxicity and it could be associated with UQCRB inhibition in the nutrient starvation condition (Page 4, Line 130-136).

#8. Section 2.2, UQCRB MT promotes the mROS level, and MT had higher LC3B and p62 levels. The results cannot support the relationship between mROS and autophagy induction in your system. Moreover, lacking autophagy usually leads to accumulation of p62 but not decrease. On the other hand, autophagy induction will reduce the p62 level.

Response: Thank you for your very insightful comments.

Previously, Zhang X. et al. proposed a model wherein an elevation of mitochondrial ROS levels leads to lysosomal calcium channel transient receptor potential mucolipin 1 (TRPML1) activation and lysosomal Ca2+ release [9]. This activation triggers calcineurin-dependent TFEB-nuclear translocation, autophagy induction and lysosome biogenesis. Thus we investigated this possibility in our study and confirmed that increased level of mROS production in MT cells resulted in activation of TRPML1 induced Ca2+ release and TFEB activation. TFEB is well known master gene for lysosomal biogenesis, coordinated this program by driving expression of autophagy and lysosomal genes [10]. Inhibition of UQCRB by A1938 abolished TFEB-nuclear translocation in MT and HCT116 cells. This led us to speculate that regulating mROS could reduce the autophagy activity in these cells.

SQSTM1/p62 is a multifunctional adaptor protein that recruits ubiquitinated proteins and organelles to LC3II to be taken into the autophagosome and broken down after lysosome fusion, therefore it should be break down in the end of the autophagy activity. However, Chen C. et al exhibited that lipopolysaccharide stimulates p62-dependent autophagy in hepatocytes and until 24 h treatment, p62 protein level was increased [11]. Lacking autophagy usually leads to accumulation of p62, although in some cases the increase in p62 level was observed when autophagy was activated [12].

In addition, in Supplementary Figure 1, we examined mROS generation in HEK293 and MT cells by using the mROS-specific red fluorescence indicator MitoSOX™. Like the previous studies’ result, mROS generation was increased in MT cells compared to control cells. In Figure 2E, LC3B-II:LC3B-I ratio and p62 protein level is increased in MT cells. When CQ treated, it shows the large difference between HEK293 and MT cells. Therefore, we thought that the results and references can support the relationship between mROS and autophagy induction in our system.

<Reference>

  1. Zhang, X.; Cheng, X.; Yu, L.; Yang, J.; Calvo, R.; Patnaik, S.; Hu, X.; Gao, Q.; Yang, M.; Lawas, M. MCOLN1 is a ROS sensor in lysosomes that regulates autophagy. Nature communications 2016, 7, 1-12.
  2. Settembre, C.; Di Malta, C.; Polito, V.A.; Arencibia, M.G.; Vetrini, F.; Erdin, S.; Erdin, S.U.; Huynh, T.; Medina, D.; Colella, P. TFEB links autophagy to lysosomal biogenesis. science 2011, 332, 1429-1433.
  3. Chen, C.; Deng, M.; Sun, Q.; Loughran, P.; Billiar, T.R.; Scott, M.J. Lipopolysaccharide stimulates p62-dependent autophagy-like aggregate clearance in hepatocytes. BioMed research international 2014, 2014.
  4. Hwang, H.-Y.; Shim, J.S.; Kim, D.; Kwon, H.J. Antidepressant drug sertraline modulates AMPK-MTOR signaling-mediated autophagy via targeting mitochondrial VDAC1 protein. Autophagy 2021, 17, 2783-2799.

#9. All the GFP-LC3 or RFP-LC3 figures should clearly show the LC3 puncta.

Response: Thank you for pointing out this. We provide other high resolution images for Figure 2F.

#10. Can the author exclude those green dots that are not dead cells?

Response: Thank you for pointing out this. We have tried to exclude green dots in Figures, however, it was hard to observe the GFP-LC3 puncta from the representative image when we decreased the intensity value. We will try to get better cell images in coming investigations.

#11. The quality of Figure 4D is poor. Please improve it and add the statistic result in Figure 4 E.

Response: Thank you for pointing out this. We revised the western blot result into the earlier expose time image and added the statistic result in Figure 4E. Although we could not confirm the meaningful statistic result after treatment of A1938 in MT cells in western blot assay, in Figure 4A and B showed that A1938 treatment can reduce acidic lysosome vesicle significantly. In this context, it would be appropriate to have the 2.4. section’s title as ‘The effect of A1938 on decreased active-lysosomes in MT cells’.

#12. Section 2.5.” MT cells treated with A1938 exhibited in-creased LC3B-II and p62 protein levels’ however, the conclusion in this part is “….which indicates that A1938 decreased autophagy flux.’ Can you explain why? Moreover, A1938 increases LC3B-II in figure 5A but decease LC3B-II in figure 5D in the same MT cells. Why?

Response: Thank you for your valuable comments and we apologize the reviewer for the confusion. The result of Figure 4 provided the effect of A1938 on impairing lysosome activity in MT cells. In Figure 5B and 5C, in cells expressing mRFP-GFP-LC3, A1938 treatment resulted in decreased red fluorescence in MT cells, indicating that autolysosome is reduced and autophagy is inhibited.

Indeed we made a mistake to misinterpretate this Figure 5. We have observed the decrease in LC3B-II level by A1938 in Figure 5D, however, it was 6 h treatment in medium containing 10% serum. Since it is not consistent with the study setting, we exclude this data from the revised manuscript accordingly.

#13. In figure 6C, the author again showed that A1938 dramatically increased LC3B-II. As we know, salvation alone also can induce autophagic flux. The author needs to add control to get a solid conclusion.

Response: We appreciate the reviewer for the valuable comments that starvation alone can activate autophagic flux. In this study, we demonstrated that nutrient-starvation, which mimics the tumor environment, further enhanced the susceptibility of UQCRB overexpressed cells to cell death by regulation of the mROS level. If the increasing level of LC3B-II level is due to the starvation, LC3B-II level should remain unchanged level in every sample that we showed in Figure 6C. However, even in the same starvation situation, HCT116 cells treated with A1938 exhibited increased LC3B-II protein level, suggesting that increasing levels of mROS caused by overexpression of UQCRB in human colon carcinoma HCT116 cells could be linked to autophagy for cell survival.

In addition to the revisions commented in the responses, minor typos and errors in statements were corrected throughout the manuscript.

I believe that the revised version of manuscript is now substantially improved to meet the standard of the Journal. I would be grateful if this revised manuscript could be reviewed again and considered for publication in “Cancers.

Thank you in advance for your kind consideration on this manuscript.

Sincerely,

Ho Jeong Kwon, Ph.D., Professor

Director, Chemical Genomics Laboratory,

Department of Biotechnology, Yonsei University, Seoul 03722, Korea.

Tel: 82-2-2123-5883, Fax: 82-2-362-7265, e-mail: [email protected]

Reviewer 2 Report

Decision: Reject

The manuscript titled “Mitochondrial ROS Produced in Human Colon Carcinoma

HCT116 Cells Reduces Cell Survival via Autophagy” investigate the molecular mechanisms linking increased production of mitochondrial ROS (mROS) and enhanced autophagic flux. In human colon carcinomas cell HCT116, the authors observed increased expression of Ubiquinol cytochrome c reductase binding protein (UQCRB), which plays an essential role in transporting electrons and maintaining the structure of complex III of the electron transport chain in the mitochondria. Using a UQCRB inhibitor A1938, the authors demonstrated that UQCRB activated lysosomal TRPML1 channels leading to the release of Ca2+, which further activated TFEB resulted in elevated autophagic flux, which renders survival advantage of the cancer cells. However, the conclusions were not supported by solid evidence. Further, the manuscript is not well prepared. Comments were provided for consideration:

  1. Please double-check the title, which is the opposite of the results.
  2. Please provide more background information for the MT cells.
  3. Supplemental figures are missing.
  4. The majority of the experiments were conducted in HEK cells, which are not human colon carcinoma cells.
  5. Please include the catalog # for the antibodies.
  6. Please include line # in the manuscript for the sake of the reviewers.
  7. The conclusions need to be verified in other colon carcinoma cell lines.

Author Response

February 9, 2022

Dear Prof. Dr. Samuel C. Mok, Editor-in-Chief, Cancers

CC: Prof. Dr. Gyorgy Marko-Varga, Guest editor, Cancers

    Ms. Lillian Cao, Assistant Editor, Cancers

We are grateful to reviewers for their time and valuable comments on our manuscript entitled “Mitochondrial ROS Produced in Human Colon Carcinoma HCT116 Cells Reduces Cell Survival via Autophagy” (cancers-1585152). We have fully addressed reviewers’ concerns and extensively revised the manuscript accordingly. Followings are point-to-point responses to comments by the reviewers.

  • Revised portions were highlighted in red in the revised manuscript.

Reviewers' comments:

Reviewer #2

The manuscript titled “Mitochondrial ROS Produced in Human Colon Carcinoma HCT116 Cells Reduces Cell Survival via Autophagy” investigate the molecular mechanisms linking increased production of mitochondrial ROS (mROS) and enhanced autophagic flux. In human colon carcinomas cell HCT116, the authors observed increased expression of Ubiquinol cytochrome c reductase binding protein (UQCRB), which plays an essential role in transporting electrons and maintaining the structure of complex III of the electron transport chain in the mitochondria. Using a UQCRB inhibitor A1938, the authors demonstrated that UQCRB activated lysosomal TRPML1 channels leading to the release of Ca2+, which further activated TFEB resulted in elevated autophagic flux, which renders survival advantage of the cancer cells. However, the conclusions were not supported by solid evidence. Further, the manuscript is not well prepared. Comments were provided for consideration:

#1. Please double-check the title, which is the opposite of the results.

Response: Thank you for valuable comments. According to your’s and reviewer 3’s recommendations, we would like to change the title to ‘Mitochondrial ROS Produced in Human Colon Carcinoma HCT116 Cells Associated with Cell Survival via Autophagy’. In addition, the typos in Figure 5 title and Figure 6 title are corrected throughout the revised manuscript accordingly.

#2. Please provide more background information for the MT cells.

Response: Thank you for pointing out this. We added MT cells information Introduction part (Page 2, Line 55-65) and Materials and Methods (Page 13, Line 421-424) part. Described below is our revised manuscript accordingly.

à UQCRB mutation can cause mitochondrial defects and is associated with several diseases. A case report of a Turkish female infant showed that 4-bp deletion in the UQCRB encoding gene resulted in metabolic disorders, such as hypoglycemia and lactic acidosis [13]. In previous study, we constructed a mutant UQCRB-expressing stable cell line. Notably, MT cells exhibit more cell growth and pro-angiogenesis activities than the wild type of HEK293 cells [14]. In addition, MT cells’ mitochondria showed morphological abnormalities and they were more sensitive to UQCRB inhibitors terpestacin and A1938 than HEK293 [4,14]. These results collectively show that the redox stress mechanism operates through increasing mROS generation can cause pseudo-hypoxia. Thus, it may link mitochondrial abnormalities to angiogenesis-related diseases and cancer [15].

<Reference>

  1. Hong, J.W.; Kim, J.M.; Kim, J.E.; Cho, H.; Kim, D.; Kim, W.; Oh, J.-W.; Kwon, H.J. MiR-4435 is an UQCRB-related circulating miRNA in human colorectal cancer. Scientific reports 2020, 10, 1-11.
  2. Haut, S.; Brivet, M.; Touati, G.; Rustin, P.; Lebon, S.; Garcia-Cazorla, A.; Saudubray, J.M.; Boutron, A.; Legrand, A.; Slama, A. A deletion in the human QP-C gene causes a complex III deficiency resulting in hypoglycaemia and lactic acidosis. Human genetics 2003, 113, 118-122.
  3. Chang, J.; Jung, H.J.; Jeong, S.H.; Kim, H.K.; Han, J.; Kwon, H.J. A mutation in the mitochondrial protein UQCRB promotes angiogenesis through the generation of mitochondrial reactive oxygen species. Biochemical and biophysical research communications 2014, 455, 290-297.
  4. Gottlieb, E.; Tomlinson, I.P. Mitochondrial tumour suppressors: a genetic and biochemical update. Nature Reviews Cancer 2005, 5, 857-866.

#3. Supplemental figures are missing.

Response: We have submitted supplemental figures before, however, it might be mistakenly did not notice. The supplemental figures are resubmitted as a separate file in the journal submission system.

#4. The majority of the experiments were conducted in HEK cells, which are not human colon carcinoma cells.

Response: This will be addressed together with the comment #7 below. 

#5. Please include the catalog # for the antibodies.

Response: Thank you for pointing out this. We provide the information accordingly (Page 13, Line 436-438).

#6. Please include line # in the manuscript for the sake of the reviewers.

Response: We apologize for the inconvenience and the line number is added in our revised manuscript.

#7. The conclusions need to be verified in other colon carcinoma cell lines.

Response: Thank you for your critical comment. As this paper focused on mechanism studies of autophagy inhibitory effect by regulating UQCRB, we focused to validate this relationship between wild type of HEK293 cells and mutant UQCRB-expressing stable cell line. However, since cancer is a heterogeneous disease, diverse CRC cell lines is needed to be investigated to demonstrate its biological evaluation robustly. In the line of this, human colorectal cancer cells, HCT116, was selected to address this point in this study. In addition, we agree with your points and will do the follow-up studies to evaluate the effect of UQCRB inhibitor on diverse UQCRB overexpression cancer cell lines.

In addition to the revisions commented in the responses, minor typos and errors in statements were corrected throughout the manuscript.

I believe that the revised version of manuscript is now substantially improved to meet the standard of the Journal. I would be grateful if this revised manuscript could be reviewed again and considered for publication in “Cancers.

Thank you in advance for your kind consideration on this manuscript.

Sincerely,

Ho Jeong Kwon, Ph.D., Professor

Director, Chemical Genomics Laboratory,

Department of Biotechnology, Yonsei University, Seoul 03722, Korea.

Tel: 82-2-2123-5883, Fax: 82-2-362-7265, e-mail: [email protected]

Reviewer 3 Report

This is an interesting story and brings in novel mechanisms of autophagy and lysosome regulation in cancer cells. 

The conclusions are stated accurately in the conclusion section of the manuscript,  however the manuscript title and Figure 5 legend title represents an over-interpretation of the results.  Essentially the authors demonstrated an association between the mitochondrial ROS (mROS) production and cell survival via autophagy, but they did not prove a direct link.  They showed that inhibition of UQCRB increased mROS and decreased culture growth, while induction of  autophagy via serum-free conditions further reduced viability.   To test if this this autophagy served as a survival function, the authors show that CQ block of autophagic flux under serum free conditions further decreased survival (Figure 1).  They also show in Fig 5 that UQCRB (it is not clear if this is wild type or mutant - see below) expression inhibits autophagy and that CQ can prevent this autophagy inhibition, but it does not appear that mROS is being measured in this figure, unless perhaps panels E and F are mislabeled?  Thus, they should revise the manuscript to change to title stating "associated with" instead of "reduces" or prove that suppressing mROS (e.g. using an antioxidant - perhaps this is what mTP is and Fig 5E and F are measuring ROS?- see below) also affects the cell survival via autophagy.

The data is not interpretable unless the authors state what "mTP" is. 

Whether or not A1938 is the same or different than terpestacin should be stated the first time A1938 is written in a sentence of the manuscript.

The fact that high micromolar levels of A1938 are needed should be included in the discussion or conclusions when the value of UQCRB as a drug target and the potential of UQCRB-targeted drugs are discussed.  The need for A1938 analogs with increased potency could be stated.

In the third paragraph of the results section, the sentence  "Cell viability significantly decreased under serum-starved conditions ..." should be supported by statistical analysis to justify the word "significantly".  Also, the effects of CQ should be added to this sentence or another sentence added afterword to add the CQ results to the conclusion.

The effect of the PTD-conjugation on the UQCRB protein should be described.  Whether or not the UQCRB mutant used in this manuscript is the same or different than the PTD-UQCRB mutant should be stated.  If the latter is true, the mutant and its altered protein function should be described. 

This study is limited to one cancer and one non-cancer cell line.  The key experiments should be repeated with at least on additional cancer cell line to verify that the results are not just do to differences in the two cell lines.

Author Response

February 9, 2022

Dear Prof. Dr. Samuel C. Mok, Editor-in-Chief, Cancers

CC: Prof. Dr. Gyorgy Marko-Varga, Guest editor, Cancers

    Ms. Lillian Cao, Assistant Editor, Cancers

We are grateful to reviewers for their time and valuable comments on our manuscript entitled “Mitochondrial ROS Produced in Human Colon Carcinoma HCT116 Cells Reduces Cell Survival via Autophagy” (cancers-1585152). We have fully addressed reviewers’ concerns and extensively revised the manuscript accordingly. Followings are point-to-point responses to comments by the reviewers.

  • Revised portions were highlighted in red in the revised manuscript.

Reviewers' comments:

Reviewer #3

This is an interesting story and brings in novel mechanisms of autophagy and lysosome regulation in cancer cells.

The conclusions are stated accurately in the conclusion section of the manuscript, however the manuscript title and Figure 5 legend title represents an over-interpretation of the results. Essentially the authors demonstrated an association between the mitochondrial ROS (mROS) production and cell survival via autophagy, but they did not prove a direct link. They showed that inhibition of UQCRB increased mROS and decreased culture growth, while induction of autophagy via serum-free conditions further reduced viability. To test if this this autophagy served as a survival function, the authors show that CQ block of autophagic flux under serum free conditions further decreased survival (Figure 1).  They also show in Fig 5 that UQCRB (it is not clear if this is wild type or mutant - see below) expression inhibits autophagy and that CQ can prevent this autophagy inhibition, but it does not appear that mROS is being measured in this figure, unless perhaps panels E and F are mislabeled?  Thus, they should revise the manuscript to change to title stating "associated with" instead of "reduces" or prove that suppressing mROS (e.g. using an antioxidant - perhaps this is what mTP is and Fig 5E and F are measuring ROS?- see below) also affects the cell survival via autophagy.

 #1. The data is not interpretable unless the authors state what "mTP" is.

Response: We are grateful to the reviewer to point out the important issue that we did not fully provide an explanation behind the result. Mito-Tempo (mTP) is a mitochondria targeted antioxidant with superoxide and alkyl radical scavenging properties [16]. We used this antioxidant for removing mitochondrial ROS (mROS) specifically so that this impairing autophagy activity is caused by reduction of mROS level in UQCRB overexpressed cells.

<Reference>

  1. Ni, R.; Cao, T.; Xiong, S.; Ma, J.; Fan, G.-C.; Lacefield, J.C.; Lu, Y.; Le Tissier, S.; Peng, T. Therapeutic inhibition of mitochondrial reactive oxygen species with mito-TEMPO reduces diabetic cardiomyopathy. Free Radical Biology and Medicine 2016, 90, 12-23.

#2. Whether or not A1938 is the same or different than terpestacin should be stated the first time A1938 is written in a sentence of the manuscript.

Response: Thank you for pointing out this and we apologize the reviewer for the confusion. We added detailed description of terpestacin and A1938 in the introduction part. Described below is our revised part of manuscript.

à In our previous study, we discovered UQCRB as a target protein of terpestacin, a small molecule that inhibits angiogenic responses. Terpestacin is a sesterterpene natural product that is produced by Embellisia chlamydospora [17]. Based on this result, we studied a target-based screen with structural information on the binding mode of terpestacin and UQCRB. Subsequently, the synthetic compound, A1938 was generated and identified to exhibit potent antiangiogenic activity both in vitro and in vivo by directly binding to UQCRB [18] (Page 2-3, Line 66-72).

<Reference>

  1. Jung, H.J.; Shim, J.S.; Lee, J.; Song, Y.M.; Park, K.C.; Choi, S.H.; Kim, N.D.; Yoon, J.H.; Mungai, P.T.; Schumacker, P.T. Terpestacin inhibits tumor angiogenesis by targeting UQCRB of mitochondrial complex III and suppressing hypoxia-induced reactive oxygen species production and cellular oxygen sensing. Journal of Biological Chemistry 2010, 285, 11584-11595.
  2. Jung, H.J.; Cho, M.; Kim, Y.; Han, G.; Kwon, H.J. Development of a Novel Class of Mitochondrial Ubiquinol–Cytochrome c Reductase Binding Protein (UQCRB) Modulators as Promising Antiangiogenic Leads. Journal of medicinal chemistry 2014, 57, 7990-7998.

 #3. The fact that high micromolar levels of A1938 are needed should be included in the discussion or conclusions when the value of UQCRB as a drug target and the potential of UQCRB-targeted drugs are discussed. The need for A1938 analogs with increased potency could be stated.

Response: We appreciate the reviewer for the constructive comments. We totally agree with your concerns and added the description in the conclusion part of the necessity of improving A1938’s pharmacological properties for the potential therapeutic value to treat CRC, particularly overexpressing UQCRB protein (Page 15, Line 528-531).

#4. In the third paragraph of the results section, the sentence  "Cell viability significantly decreased under serum-starved conditions ..." should be supported by statistical analysis to justify the word "significantly".  Also, the effects of CQ should be added to this sentence or another sentence added afterword to add the CQ results to the conclusion.

Response: We are grateful to the reviewer to point out the important issue that we did not fully provide an explanation behind the wording ‘significantly’. In figure 1E, when HCT116 cells treated with 30 μM A1938 in medium containing 10% serum, cells viability was about 90%. However, at doses of 30 μM under serum-starved conditions, cells viability was decreased below 13%. From the reason, we used the word ‘significantly’. However, to interpret our data more objectively, we revised our previous description of this part in the revised manuscript accordingly.

Described below is our revised interpretation and the conclusion:

à At doses above 20 μM under serum-starved conditions, A1938 had a cytotoxic effect on HCT116 cells of over 67%. In addition, when HCT116 cells treated with 30 μM A1938 in medium containing 10% serum, cells viability was about 90%. However, at doses of 30 μM under serum-starved conditions, cells viability was decreased below 13% (Fig. 1E). To determine if the decreased cell growth and viability were the results of autophagy inhibition, HCT116 cells were treated with the autophagy inhibitor, chloroquine (CQ) (Fig. 1F). Co-treatment of CQ and A1938 shows further reduce cell viability indicating that autophagy inhibition by CQ and A1938 could play an additive role in these cells. Increased autophagy activation in HCT116 cells involving recycling of cell components to facilitate survival in a nutrient deprived environment. However, inhibition of autophagy flux in HCT116 cells in serum starvation condition would make these cell could not afford the nutrient by recycling of cell components. Taken together, these results suggested that inhibiting autophagy flux by CQ in HCT116 cells led to further increased cytotoxicity and it could be associated with UQCRB inhibition in the nutrient starvation condition (Page 4, Line 118-132).

#5. The effect of the PTD-conjugation on the UQCRB protein should be described.  Whether or not the UQCRB mutant used in this manuscript is the same or different than the PTD-UQCRB mutant should be stated.  If the latter is true, the mutant and its altered protein function should be described.

Response: Thank you for pointing out this. To investigate the biological role of UQCRB, we referred to a UQCRB mutant identified previously in human cDNA and established the mutant UQCRB-overexpressing cell line (MT) previously by our group. This mutant was cloned with a 4-bp deletion at nucleotides 338–341 of the UQCRB gene, based on the human case report, and subcloned for protein expression in mammalian cells. The resultant UQCRB mutant protein had alterations in seven amino acid residues and an additional stretch of 14 amino acids at the C-terminal end [14]. So this MT cells are overexpressing the mutant UQCRB protein compare to the normal HEK293 cells.

 On the other hand, we have generated a novel protein transduction domain (PTD)-conjugated UQCRB fusion protein before. It was originally made verifying the potential role of UQCRB as an angiogenesis enhancer. The full length of wild type UQCRB gene was subcloned into the pRSET-B protein expression vector, which contained the protein transduction domain (PTD), Hph-1 [19].

From these individual studies, we have confirmed that either mutant UQCRB overexpressing condition or wild UQCRB overexpressing condition in cells could increase the level of mROS. And it promoted us to expand our studies to new therapeutic approached for human cancer which naturally overexpress UQCRB.

<Reference>

  1. Chang, J.; Jung, H.J.; Jeong, S.H.; Kim, H.K.; Han, J.; Kwon, H.J. A mutation in the mitochondrial protein UQCRB promotes angiogenesis through the generation of mitochondrial reactive oxygen species. Biochemical and biophysical research communications 2014, 455, 290-297.
  2. Chang, J.; Jung, H.J.; Park, H.-J.; Cho, S.-W.; Lee, S.-K.; Kwon, H.J. Cell-permeable mitochondrial ubiquinol–cytochrome c reductase binding protein induces angiogenesis in vitro and in vivo. Cancer letters 2015, 366, 52-60.

#6. This study is limited to one cancer and one non-cancer cell line.  The key experiments should be repeated with at least on additional cancer cell line to verify that the results are not just do to differences in the two cell lines.

Response: Thank you for your valuable comment. As this paper focused on mechanism studies of autophagy inhibitory effect by regulating UQCRB, we focused to validate this relationship between wild type of HEK293 cells and mutant UQCRB-expressing stable cell line. However, since cancer is a heterogeneous disease, diverse CRC cell lines is needed to be investigated to demonstrate its biological evaluation robustly. In the line of this, human colorectal cancer cells, HCT116, was selected to address this point in this study. In addition, we agree with your points and will  the follow-up studies to evaluate the effect of UQCRB inhibitor on diverse UQCRB overexpression cancer cell lines.

In addition to the revisions commented in the responses, minor typos and errors in statements were corrected throughout the manuscript.

I believe that the revised version of manuscript is now substantially improved to meet the standard of the Journal. I would be grateful if this revised manuscript could be reviewed again and considered for publication in “Cancers.

Thank you in advance for your kind consideration on this manuscript.

Sincerely,

Ho Jeong Kwon, Ph.D., Professor

Director, Chemical Genomics Laboratory,

Department of Biotechnology, Yonsei University, Seoul 03722, Korea.

Tel: 82-2-2123-5883, Fax: 82-2-362-7265, e-mail: [email protected]

Reviewer 4 Report

In their paper titled: "Mitochondrial ROS Produced in Human Colon Carcinoma HCT116 Cells Reduces Cell Survival via Autophagy" the authors investigated the role of UQCRB and A1938 in autophagy, cell death and proliferation. The work is interesting, and the experiments are appropriately selected. However, some points should be reviewed before acceptance.

- Abbreviations should not be in the abstract (TRPML1, EB);

- I would expect a characterization of A1938 as the molecule studied in the introduction.

- Formazan is formed in mitochondria during the MTT assay. Under certain conditions, the reaction responsible for formazan production is catalysed (antioxidant), but cell number does not change. The MTT assay relates to the metabolic activity of cells rather than cell growth.

- Page 3: "HCT116 cells were treated with the autophagy inhibitor chloroquine (CQ) (Fig. 1D)." It should be Fig. 1F.

- Figure 1E: What are S-10 media and SF media?

- There is a significant discrepancy between Figures 1D and 1E. In Figure 1E, cell viability at 50 µM A1938 is almost 100%, but in Figure 1D there is a 50% difference.

- Regarding the statement, "Cell viability significantly decreased under serum starvation conditions as a result of increased autophagy activation involving recycling of cell components to facilitate survival in a nutrient deprived environment." I see a discrepancy in Figure 1E and 1F. Chloroquine was used as an inhibitor of autophagy. If the absence of serum induces autophagy, then cell viability should be at the same level when treated with chloroquine and A1938 as in Figure 1E. The authors should describe the last paragraph on page 3 better and more carefully. Perhaps pay more attention to the explanation of autophagy activation/inhibition, autophagosome formation, and autophagy flux.

- I suggest replacing Figure 2 with more representative images. Only one out of five cells expresses GFP and RFP (according to nuclear staining).

- Define D.W. in the (G) legend of Figure 2.

- The abbreviation TRPML1 is not explained.

- What is mTP in Figure 5?

Author Response

February 9, 2022

Dear Prof. Dr. Samuel C. Mok, Editor-in-Chief, Cancers

CC: Prof. Dr. Gyorgy Marko-Varga, Guest editor, Cancers

    Ms. Lillian Cao, Assistant Editor, Cancers

We are grateful to reviewers for their time and valuable comments on our manuscript entitled “Mitochondrial ROS Produced in Human Colon Carcinoma HCT116 Cells Reduces Cell Survival via Autophagy” (cancers-1585152). We have fully addressed reviewers’ concerns and extensively revised the manuscript accordingly. Followings are point-to-point responses to comments by the reviewers.

  • Revised portions were highlighted in red in the revised manuscript.

Reviewers' comments:

Reviewer #4

In their paper titled: "Mitochondrial ROS Produced in Human Colon Carcinoma HCT116 Cells Reduces Cell Survival via Autophagy" the authors investigated the role of UQCRB and A1938 in autophagy, cell death and proliferation. The work is interesting, and the experiments are appropriately selected. However, some points should be reviewed before acceptance.

#1. Abbreviations should not be in the abstract (TRPML1, EB);

Response: Thank you for pointing out these errors. ‘TRPML1’ has been corrected as full name which is ‘Transient receptor potential mucolipin 1’. However, ‘EB’ is the full name of EB is the ‘Transcription factor EB’.

#2. I would expect a characterization of A1938 as the molecule studied in the introduction.

Response: Thank you for pointing out this. We added detailed description of A1938 in the introduction part. Described below is our revised part of manuscript.

à In our previous study, we discovered UQCRB as a target protein of terpestacin, a small molecule that inhibits angiogenic responses. Terpestacin is a sesterterpene natural product that is produced by Embellisia chlamydospora [17]. Based on this result, we studied a target-based screen with structural information on the binding mode of terpestacin and UQCRB. Subsequently, the synthetic compound, A1938 was generated and identified to exhibit potent antiangiogenic activity both in vitro and in vivo by directly binding to UQCRB [18] (Page 2-3, Line 66-72).

<Reference>

  1. Jung, H.J.; Shim, J.S.; Lee, J.; Song, Y.M.; Park, K.C.; Choi, S.H.; Kim, N.D.; Yoon, J.H.; Mungai, P.T.; Schumacker, P.T. Terpestacin inhibits tumor angiogenesis by targeting UQCRB of mitochondrial complex III and suppressing hypoxia-induced reactive oxygen species production and cellular oxygen sensing. Journal of Biological Chemistry 2010, 285, 11584-11595.
  2. Jung, H.J.; Cho, M.; Kim, Y.; Han, G.; Kwon, H.J. Development of a Novel Class of Mitochondrial Ubiquinol–Cytochrome c Reductase Binding Protein (UQCRB) Modulators as Promising Antiangiogenic Leads. Journal of medicinal chemistry 2014, 57, 7990-7998.

#3. Formazan is formed in mitochondria during the MTT assay. Under certain conditions, the reaction responsible for formazan production is catalysed (antioxidant), but cell number does not change. The MTT assay relates to the metabolic activity of cells rather than cell growth.

Response: Thank you for your valuable comment. MTT assay could use for the metabolic activity, however, MTT assay has been introduced as suitable method to measure cell proliferation as shown in many previous reports [20-22]. Accordingly, we used MTT assay for investigation of cell proliferation and showed A1938 could decrease cell proliferation rate in a dose-dependent manner. However, we do appreciate your valuable concern on MTT assay and will look for other assay to investigate the cell growth.

<Reference>

  1. Cho, S.M.; Lee, H.J.; Karuso, P.; Kwon, H.J. Daptomycin suppresses tumor migration and angiogenesis via binding to ribosomal protein S19 in humans. The Journal of Antibiotics 2021, 74, 726-733.
  2. Choi, I.-K.; Cho, Y.S.; Jung, H.J.; Kwon, H.J. Autophagonizer, a novel synthetic small molecule, induces autophagic cell death. Biochemical and biophysical research communications 2010, 393, 849-854.
  3. Weichert, H.; Blechschmidt, I.; Schröder, S.; Ambrosius, H. The MTT-assay as a rapid test for cell proliferation and cell killing: application to human peripheral blood lymphocytes (PBL). Allergie und Immunologie 1991, 37, 139-144.

#4. Page 3: "HCT116 cells were treated with the autophagy inhibitor chloroquine (CQ) (Fig. 1D)." It should be Fig. 1F.

Response: Thank you for pointing out these errors. The typos are corrected throughout the revised manuscript accordingly.

#5. Figure 1E: What are S-10 media and SF media?

Response: Thank you for pointing out this and we apologize the reviewer for the confusion. S-10 media is DMEM (or RPMI) supplemented with 10% FBS and 1% antibiotics and SF media is DMEM (or RPMI) supplemented with 1% antibiotics. We used SF media for establishment of the serum-starvation condition. According to your comment, we added explanation in Figure 1’s legend. Described below is our revised part of manuscript.

à S-10 media means medium containing 10% serum and SF media does medium without the serum. SF media have been used for establishing the serum-starvation conditions.

#6. There is a significant discrepancy between Figures 1D and 1E. In Figure 1E, cell viability at 50 µM A1938 is almost 100%, but in Figure 1D there is a 50% difference.

Response: Thank you for pointing out this. In Figure 1D’s result, cells’ viability is measured by O.D. (595 nm) so please compare the data from the 0 h O.D. value for the cell viability. 72 h of 50 μM A1938 treatment O.D. value is about 0.45 and 0 h O.D. value was about 0.3. So in this case, we speculated that 50 μM A1938 treatment made increased cell proliferation. Thus it is hard to say Figure 1D’s viability at 50 μM A1938 is about 50% (Please check the explanation in the figure right).

 #7. Regarding the statement, "Cell viability significantly decreased under serum starvation conditions as a result of increased autophagy activation involving recycling of cell components to facilitate survival in a nutrient deprived environment." I see a discrepancy in Figure 1E and 1F. Chloroquine was used as an inhibitor of autophagy. If the absence of serum induces autophagy, then cell viability should be at the same level when treated with chloroquine and A1938 as in Figure 1E. The authors should describe the last paragraph on page 3 better and more carefully. Perhaps pay more attention to the explanation of autophagy activation/inhibition, autophagosome formation, and autophagy flux.

Response: We are grateful to the reviewer to point out the important issue that we did not fully provide an explanation behind these result. In the case of solid tumors, autophagy capability is often increased to escape nutrient deprivation [23]. Since CRC is one of the most frequent solid tumor, in Figure E, cell viability test was conducted in the presence or absence of serum for mimic the native tumor environment. In addition, in Figure E, not CQ but A1938 was treated as variety of dose respectively. In Figure F, we used CQ as an autophagy inhibitor by impairing autophagosome fusion with lysosome. We thought that A1938 could have the potential to affect the growth of HCT116 cells due to its suppression of mROS production. As described earlier, Zhang X. et al. suggested a model wherein an elevation of mitochondrial ROS levels leads to lysosomal calcium channel transient receptor potential mucolipin 1 (TRPML1) activation and lysosomal Ca2+ release [9]. This activation triggers calcineurin-dependent TFEB-nuclear translocation, autophagy induction and lysosome biogenesis. In this context, A1938 treatment in the presence or absence of CQ have been conducted. By Figure 1F, we observed further reduce cell viability while CQ and A1938 was co-treated which indicates that autophagy inhibition by CQ and A1938 could play an additive role in these cells.

However, as we agree to the reviewer’s point that we have to pay more attention to the explanation in last paragraph on page 3, we revised interpretation of the observation. Described below is our revised part of manuscript.

à At doses above 20 μM under serum-starved conditions, A1938 had a cytotoxic effect on HCT116 cells of over 67%. In addition, when HCT116 cells treated with 30 μM A1938 in medium containing 10% serum, cells viability was about 90%. However, at doses of 30 μM under serum-starved conditions, cells viability was decreased below 13% (Fig. 1E). To determine if the decreased cell growth and viability were the results of autophagy inhibition, HCT116 cells were treated with the autophagy inhibitor, chloroquine (CQ) (Fig. 1F). Co-treatment of CQ and A1938 shows further reduction of cell viability indicating that autophagy inhibition by CQ and A1938 could play an additive role in these cells. Increased autophagy activation in HCT116 cells involving recycling of cell components to facilitate survival in a nutrient deprived environment. However, inhibition of autophagy flux in HCT116 cells in serum starvation condition would make these cell could not afford the nutrient by recycling of cell components. Taken together, these results suggested that inhibiting autophagy flux by CQ in HCT116 cells led to further increased cytotoxicity and it could be associated with UQCRB inhibition in the nutrient starvation condition (Page 4, Line 118-132).

<Reference>

  1. Zhang, X.; Cheng, X.; Yu, L.; Yang, J.; Calvo, R.; Patnaik, S.; Hu, X.; Gao, Q.; Yang, M.; Lawas, M. MCOLN1 is a ROS sensor in lysosomes that regulates autophagy. Nature communications 2016, 7, 1-12.
  2. Fu, Y.; Hong, L.; Xu, J.; Zhong, G.; Gu, Q.; Gu, Q.; Guan, Y.; Zheng, X.; Dai, Q.; Luo, X. Discovery of a small molecule targeting autophagy via ATG4B inhibition and cell death of colorectal cancer cells in vitro and in vivo. Autophagy 2019, 15, 295-311.

#8. I suggest replacing Figure 2 with more representative images. Only one out of five cells expresses GFP and RFP (according to nuclear staining).

Response: Thank you for pointing out this. We provide other representative images for Figure 2F.

#9. Define D.W. in the (G) legend of Figure 2.

Response: Thank you for pointing out this error. The typo is corrected to deionized water (DIW).

#10. The abbreviation TRPML1 is not explained.

Response: Thank you for pointing out this error. This abbreviation is added in abstract part.

#11. What is mTP in Figure 5?

Response: We are grateful to the reviewer to point out this. Mito-Tempo (mTP) is a mitochondria targeted antioxidant with superoxide and alkyl radical scavenging properties [16]. We used this antioxidant for removing mitochondrial ROS (mROS) specifically so that we could speculate this impairing autophagy activity is caused by reduction of mROS level in UQCRB overexpressed cells.

<Reference>

  1. Ni, R.; Cao, T.; Xiong, S.; Ma, J.; Fan, G.-C.; Lacefield, J.C.; Lu, Y.; Le Tissier, S.; Peng, T. Therapeutic inhibition of mitochondrial reactive oxygen species with mito-TEMPO reduces diabetic cardiomyopathy. Free radical biology and medicine 2016, 90, 12-23.

 In addition to the revisions commented in the responses, minor typos and errors in statements were corrected throughout the manuscript.

I believe that the revised version of manuscript is now substantially improved to meet the standard of the Journal. I would be grateful if this revised manuscript could be reviewed again and considered for publication in “Cancers.

Thank you in advance for your kind consideration on this manuscript.

Sincerely,

Ho Jeong Kwon, Ph.D., Professor

Director, Chemical Genomics Laboratory,

Department of Biotechnology, Yonsei University, Seoul 03722, Korea.

Tel: 82-2-2123-5883, Fax: 82-2-362-7265, e-mail: [email protected]

Round 2

Reviewer 1 Report

Thanks for the response and the correction.

Reviewer 2 Report

Decision: Reject
I remain my original decision, although the quality of the revised manuscript has been improved. First, the result of the in vivo study is not significant, more animals could be included in the future study to support the conclusions. Again, it’s a severe flaw of the experiment design that the authors used Human embryonic kidney (HEK) cells to study an observation found in human colon carcinoma cells without further validation in the colon carcinoma cells.  Therefore, it could be artificial observation only exist in certain in vitro study. Further, the conclusions of the whole study were based only on the UQCRB inhibitor A1938 at the relatively high dose of 30 uM, which could be problematic. Other approaches, such as shRNA or CRISPR-knockout, to inhibit UQCRB function, could be included in the future study to solidify the conclusion.

Reviewer 3 Report

The authors have edited, reorganized and clarified details sufficiently to address my questions and concerns.  The revised manuscript has very significant and interesting results and I hope that the authors continue this research utilizing additional cell lines and specific interjections in the pathways to prove the molecular and cellular effects are directly involved in the cellular outcomes.  Below are some minor concerns about the revised manuscript: 

The grammar of the following sentence on Line 63 should be corrected: “These results collectively show that the redox stress mechanism operates 63 through increasing mROS generation can cause pseudo-hypoxia should be corrected.” should be corrected.

In Figure 1F, the cut-off legend should be fixed.

In Figure 5C, the authors should specify what was compared in the statistical analysis, so that the reader can no if the mTP was significantly different from the DMSO, the A1938 or both.

The grammar of the following two sentences starting on Line 130 should be corrected: “Increased autophagy activation in HCT116 cells involving recycling of cell 130 components to facilitate survival in a nutrient deprived environment” and “However, inhibition 131 of autophagy flux in HCT116 cells in serum starvation condition would make these cell 132 could not afford the nutrient by recycling of cell components.”

Reviewer 4 Report

The authors fulfill the comments addressed by the reviewer.